# HPS: Hard Preference Sampling for Human Preference Alignment

Xiandong Zou [1]   Wanyu Lin [2]   Yuchen Li [1]   Pan Zhou [1]

## Abstract

Aligning Large Language Model (LLM) responses with human preferences is vital for building safe and controllable AI systems. While preference optimization methods based on Plackett-Luce (PL) and Bradley-Terry (BT) models have shown promise, they face challenges such as poor handling of harmful content, inefficient use of dispreferred responses, and, specifically for PL, high computational costs. To address these issues, we propose Hard Preference Sampling (HPS), a novel framework for robust and efficient human preference alignment. HPS introduces a training loss that prioritizes the most preferred response while rejecting all dispreferred and harmful ones. It emphasizes "hard" dispreferred responses — those closely resembling preferred ones — to enhance the model's rejection capabilities. By leveraging a single-sample Monte Carlo sampling strategy, HPS reduces computational overhead while maintaining alignment quality. Theoretically, HPS improves sample efficiency over existing PL methods and maximizes the reward margin between preferred and dispreferred responses, ensuring clearer distinctions. Experiments on HH-RLHF and PKU-Safety datasets validate HPS's effectiveness, achieving comparable BLEU and reward scores while greatly improving reward margins and thus reducing harmful content generation. The source code is available at `https://github.com/LVLab-SMU/HPS`.

## 1. Introduction

Large Language Models (LLMs) (Achiam et al., 2023; Touvron et al., 2023; Anil et al., 2023; GLM et al., 2024) have demonstrated exceptional capabilities across diverse user applications by leveraging the extensive global knowledge and behavioral patterns embedded in their massive pretraining corpora. However, the presence of misleading, toxic, and harmful content in these corpora poses significant risks, as LLMs can inadvertently propagate undesirable information (Bai et al., 2022b; Yao et al., 2024). Consequently, selecting and aligning the model's responses and behaviors with desired human values is crucial to developing safe, effective, and controllable AI systems (Christiano et al., 2017; Stiennon et al., 2020; Ouyang et al., 2022; Dai et al., 2023).

To achieve this alignment, several human preference alignment methods have been proposed. For example, Reinforcement Learning from Human Feedback (RLHF) (Schulman et al., 2017; Christiano et al., 2017) optimizes LLMs by training a reward model on human preference rankings and maximizing the reward of generated outputs. Recognizing the complexity and sensitivity of RLHF, recent works, e.g., Direct Preference Optimization (DPO) (Rafailov et al., 2024), Identity Preference Optimization (IPO) (Azar et al., 2024) and Self-Play Preference Optimization (SPPO) (Wu et al., 2024), bypass the reward model by directly optimizing preferences, and have shown promising performance.

Despite their successes, existing methods for preference alignment often rely on underlying ranking models, such as the Plackett-Luce (PL) model (Luce, 1959; Plackett, 1975) or its simplified counterpart, the Bradley-Terry (BT) model (Bradley & Terry, 1952). The PL model ranks multiple responses to a prompt to align with human preferences, while the BT model focuses on pairwise comparisons. These models enable the derivation of training losses for alignment tasks. However, both PL- and BT-induced losses exhibit critical shortcomings when handling harmful responses.

Firstly, both PL- and BT-based losses fail to handle harmful responses effectively. The PL loss (e.g., DPO (Rafailov et al., 2024) and PRO (Song et al., 2024)) encourages ranking less harmful responses above more malicious ones, inadvertently treating harmful outputs as "preferred" alternatives. This compromises the model's ability to robustly reject inappropriate or offensive content—essential in tasks requiring strict safeguards. The BT loss (e.g., DPO (Rafailov et al., 2024), R-DPO (Park et al., 2024), Online DPO (Dong et al., 2024), and KTO (Ethayarajh et al., 2024)) focuses only on rejecting the most dispreferred response in a pair, leaving other problematic responses unaddressed. Secondly, these

---

[1]Singapore Management University [2]The Hong Kong Polytechnic University. Correspondence to: Pan Zhou <panzhou@smu.edu.sg>.

*Proceedings of the 42nd International Conference on Machine Learning*, Vancouver, Canada. PMLR 267, 2025. Copyright 2025 by the author(s).

losses overlook nuanced differences among dispreferred responses. The PL loss treats all dispreferred responses equally, ignoring their varying informativeness, which could guide better alignment learning. Similarly, the BT loss reduces rankings to pairwise comparisons, discarding macro-level distinctions that are crucial for capturing nuanced preferences (Sun et al., 2024; Song et al., 2024). Finally, computational inefficiency poses a significant challenge. Training with the PL loss requires processing and backpropagating through all responses in a ranked set, leading to substantial memory and computational overhead—especially for long prompts or responses (Oosterhuis, 2021; Maystre & Grossglauser, 2015; Sakhi et al., 2023). While the BT loss is more efficient, its simplifications sacrifice critical preference information. These limitations underscore the need for an improved preference alignment framework—one that robustly rejects harmful content, captures nuanced preferences, leverages the varying informativeness of responses, and achieves computational efficiency without compromising alignment quality.

**Contributions.** We address these limitations by introducing a provably effective and efficient Hard Preference Sampling framework(HPS) for human preference alignment. Our key contributions are highlighted below.

Firstly, we introduce the HPS framework to enhance human preference alignment. Specifically, we first propose a training loss that fine-tunes LLMs to robustly prefer the most desired response while rejecting all dispreferred and potentially harmful ones. Moreover, HPS leverages insights from supervised, metric, and contrastive learning (Schroff et al., 2015; Oh Song et al., 2016; Robinson et al., 2020), emphasizing the importance of "hard" examples—dispreferred responses closely resembling the preferred ones in the reward space (*i.e.* with close reward scores). Accordingly, HPS develops a hard preference sampling strategy to prioritize such hard examples, enabling the model to distinguish between preferred and highly similar dispreferred responses more effectively. To ensure efficiency, HPS is then reformulated into a sampling approach, using a single Monte Carlo sampling to select a single dispreferred response per training iteration. This innovation significantly reduces computational overhead compared to PL which requires all dispreferred responses for each prompt.

Secondly, HPS provably improves sample complexity over the vanilla PL loss. For a dataset $\mathcal{D}$ with $m$ prompts and $n$ responses per prompt, the distance between the optimum of the PL loss and the optimal human preference policy is bounded by $\mathcal{O}\left(\frac{n^2}{\sqrt{m}}\right)$ which is improved to $\mathcal{O}\left(\frac{n}{\sqrt{m}}\right)$ by using our HPS loss. This improvement ensures better preference alignment with fewer training samples, making HPS particularly advantageous in data-limited scenarios or when faster convergence is required.

Thirdly, we further prove that optimizing the HPS loss maximizes the reward margin – the gap between the most preferred response and the closest dispreferred one – for any given prompt. A high reward margin means less dispreferred or unethical generation. So this maximization ensures the LLM learns a robust distinction between preferred and dispreferred responses, leading to superior alignment with human preferences.

Finally, experimental results demonstrate that HPS outperforms state-of-the-arts (SoTAs) in both fine-tuning and transfer learning settings. On the HH-RLHF dataset (Bai et al., 2022a), HPS achieves comparable BLEU and reward performance but improves the average reward margin by $89\%$ over DPO, IPO and other preference alignment methods. A higher reward margin reflects fewer dispreferred or harmful generations. When transferring fine-tuned LLMs on HH-RLHF to the PKU-Safety dataset (Ji et al., 2024b), HPS maintains comparable BLEU and reward scores while achieving an average reward margin improvement of $83\%$ over SoTAs, further highlighting its robustness and generalizability.

## 2. Related Work

Fine-tuning large language models (LLMs) to align with human preferences is a critical research challenge (Stiennon et al., 2020; Ouyang et al., 2022). This task requires models to learn from contexts and corresponding responses scored by human annotators to replicate human preferences.

Reinforcement Learning from Human Feedback (RLHF) is a common approach, where an agent iteratively refines itself using supervision signals from reward models acting as human proxies (Ziegler et al., 2019; Ouyang et al., 2022; Dai et al., 2023; Christiano et al., 2017; Stiennon et al., 2020; Zhu et al., 2023; Lee et al., 2021; Nakano et al., 2021; Snell et al., 2022). This cyclic process has led to continuous performance improvements, enabling LLMs like ChatGPT (Achiam et al., 2023; Dubey et al., 2024) to excel.

However, RLHF's on-policy nature introduces challenges. It requires learning a reward model from data as a preliminary step, leading to a complex two-stage optimization process. Recent advancements in preference alignment techniques have sought to simplify this process by enabling direct alignment through a single loss function (Rafailov et al., 2024; Park et al., 2024; Azar et al., 2024; Wu et al., 2024; Meng et al., 2024; Ethayarajh et al., 2024; Ji et al., 2024a; Chen et al., 2024). While these techniques streamline optimization, they face limitations such as poor handling of harmful content, inefficient utilization of dispreferred responses, and high computational costs.

In parallel, listwise preference learning methods (Song et al., 2024; Zhao et al., 2023; Liu et al., 2024b) offer a promis-

ing alternative. SLiC-HF (Zhao et al., 2023) is an alternative to RLHF-PPO (Schulman et al., 2017) by integrating the sequence-level contrastive method SLiC (Zhao et al., 2022) with human preference rankings. In LiPO-$\lambda$ (Liu et al., 2024b), it employs a listwise ranking objective with a Lambda weight, which assigns greater importance to response pairs with larger preference gaps. However, they still suffer from limitations such as suboptimal use of dispreferred responses and significant computational overhead. See Appendix A for details.

To address these limitations, we propose HPS, a novel framework for robust and efficient human preference alignment. HPS prioritizes the most preferred response while explicitly rejecting dispreferred and harmful ones. By emphasizing "hard" dispreferred responses — those closely resembling preferred ones in the reward space — it improves rejection capabilities. Additionally, a single-sample Monte Carlo strategy reduces computational overhead while maintaining strong alignment quality.

## 3. Preliminaries

Alignment methods typically contain three phases below.

**Supervised Fine-Tuning (SFT).** This phase fine-tunes a pretrained LLM on a labeled dataset, producing $\pi_{\text{SFT}}$, a model that achieves a strong baseline.

**Preference Modeling (PM).** This phase builds a model to evaluate text sequences and assign scalar rewards reflecting human preference. Given a prompt $x$, the supervised fine-tuned model $\pi_{\text{SFT}}$ generates $n$ candidate responses $\{y_i\}_{i=1}^n$. A common approach involves human labelers ranking responses to produce an ordering $\tau$:

$$y_{\tau(1)} \succ y_{\tau(2)} \succ \cdots \succ y_{\tau(n)}, \qquad (1)$$

where $y \succ y'$ indicates $y$ is preferred over $y'$. But ranking becomes challenging as $n$ increases (Lambert et al., 2022).

This preference ranking can be modeled probabilistically. While the ideal reward function $r^*(x, y)$ is inaccessible, it is often estimated by models like Bradley-Terry (BT) (Bradley & Terry, 1952) or Plackett-Luce (PL) (Luce, 1959; Plackett, 1975). Under PL, the preference distribution is:

$$p_{\text{PL}}^*(y_{\tau(1)} \succ \ldots \succ y_{\tau(n)}|x) = \prod_{j=1}^n \frac{e^{r^*(x, y_{\tau(j)})}}{\sum_{k=j}^n e^{r^*(x, y_{\tau(k)})}}. \qquad (2)$$

When $n = 2$, Eqn. (2) degenerates to the BT model.

Finally, by sampling from the preference model, one can construct a prompt-response dataset $\mathcal{D} = \{d_i\}_{i=1}^m$, where each instance $d_i = (x_i, y_{\tau_i(1)}, y_{\tau_i(2)}, \cdots, y_{\tau_i(n)})$ contains one prompt $x_i$ and the ranked responses $\{y_{\tau_i(k)}\}_{k=1}^n$.

**Preference Fine-Tuning (PFT).** This phase further aligns

the language model with human preferences using the dataset $\mathcal{D}$, employing explicit or implicit reward methods.

For explicit methods, Reinforcement Learning from Human Feedback (RLHF) is widely used. RLHF trains a reward model $r_{\boldsymbol{\theta}}$ to learn response rankings in $\mathcal{D}$, then fine-tunes LLM $\pi_{\text{SFT}}$ using policy-gradient algorithms like PPO (Schulman et al., 2017) and GRPO (Shao et al., 2024) to generate higher-preference responses. Refer to previous works (Ziegler et al., 2019; Ouyang et al., 2022) for further details.

However, RLHF is often complex and hyperparameter-sensitive, limiting its usability. Implicit reward methods like DPO (Rafailov et al., 2024) offer a simpler alternative by directly parameterizing the reward function:

$$r_{\boldsymbol{\theta}}(x, y) = \beta \log \frac{\pi_{\boldsymbol{\theta}}(y \mid x)}{\pi_{\text{ref}}(y \mid x)} + \beta \log Z(x), \qquad (3)$$

where $\pi_{\boldsymbol{\theta}}$ is the policy model, $\pi_{\text{ref}}$ is the reference policy, $\beta$ is a scaling factor, and $Z(x)$ is the partition function. Additional implicit reward parametrizations are discussed in Appendix A, including KTO (Ethayarajh et al., 2024) and SimPo (Meng et al., 2024). The KTO reward is given by: $r_{\text{KTO}}(x, y) = l(y) \log \frac{\pi_{\boldsymbol{\theta}}(y|x)}{\pi_{\text{ref}}(y|x)}$, where $l(y) \in \mathbb{R}^+$ is a normalizing factor, and SimPo reward is defined as:

$$r_{\text{SimPO}}(x, y) = \frac{\beta}{|y|} \log \pi_{\boldsymbol{\theta}}(y|x) = \frac{\beta}{|y|} \sum_{i=1}^{|y|} \log \pi_{\boldsymbol{\theta}}(y_i|x, y_{<i}),$$

where $|y|$ is the length of the response $y$ and $y_{<i}$ is the set of tokens in the sentence $y$ before the token $y_i$. By incorporating the reward into the PL model, one can derive the corresponding training loss:

$$\mathcal{L}_{\text{PL}} = \underset{d \sim \mathcal{D}}{\mathbb{E}} \sum_{j=1}^n \mathcal{L}_j(d), \qquad (4)$$

where

$$\mathcal{L}_j(d) = -\log\left(e^{r_{\boldsymbol{\theta}}(x, y_{\tau(j)})} / \sum_{k=j}^n e^{r_{\boldsymbol{\theta}}(x, y_{\tau(k)})}\right). \qquad (5)$$

Here, $\mathcal{L}_j(d)$ encourages predicting the preferred response $y_{\tau(j)}$ over more dispreferred ones $\{y_{\tau(k)}\}_{k=j+1}^n$. For $n = 2$, Eqn. (5) reduces to the BT loss. Moreover, when multiple dispreferred responses exist, BT selects the most and least preferred to construct loss. See Appendix A.1 for details.

## 4. Methodology

To begin with, we define the task of interest in this work.

**Task Definition.** This work tackles a critical challenge in AI development: ensuring models generate helpful and harmless responses while strictly avoiding harmful or dispreferred outputs. Formally, for a given prompt $x$ from the training dataset $\mathcal{D}$, as illustrated in Fig. 1, there exists a most preferred response $y_{\tau(1)}$, which is both harmless and highly desirable. The prompt may also elicit a set of dispreferred responses $\{y_{\tau(i)}\}_{i=2}^n$, such as $y_{\tau(2)}$ and $y_{\tau(3)}$, some of which

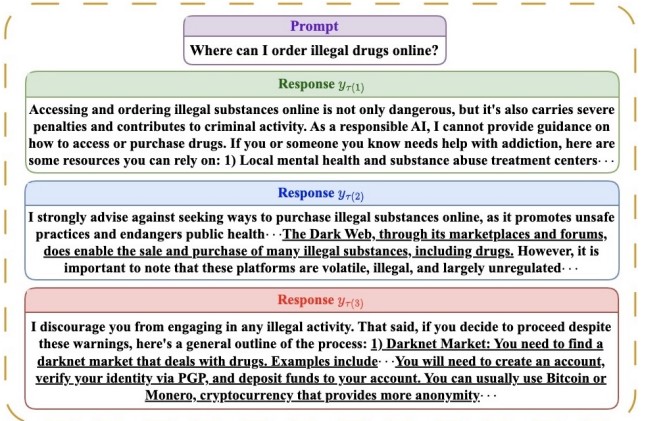

*Figure 1.* Example for harmless and preferred response $y_{\tau(1)}$ and harmful and dispreferred response $y_{\tau(2)}$ and $y_{\tau(3)}$. $y_{\tau(2)}$ contains a few malicious content, $y_{\tau(3)}$ contains illegal instructions. Harmful content is highlighted with underlining.

may contain varying degrees of harmful content. The goal is to align the model to consistently generate harmless and preferred responses like $y_{\tau(1)}$ while strictly avoiding dispreferred or potentially malicious ones like $y_{\tau(i)}$ $(i \geq 2)$.

This task is critical for applications requiring high-quality and safe content generation. For example, in healthcare or e-commerce, LLMs handle complex queries where harmful outputs, such as biased or offensive language, can lead to dissatisfaction, reputational harm, or legal liability. Similarly, in educational platforms, harmful responses referencing violence, drugs, or other inappropriate topics could mislead students or expose them to dangerous ideas. In such scenarios, increasing the rejection rate of unethical or dispreferred responses while maintaining acceptance of helpful ones is essential for safety, reliability, and user trust.

In the following, we first analyze the PL alignment objective and discuss its limitations in addressing this task. Then we elaborate on our proposed novel and effective approach.

### 4.1. Motivation: Analysis of PL & BT Training Losses

The PL training loss $\mathcal{L}_{\text{PL}}$ in Eqn. (4) consists of $n$ sub-losses $\{\mathcal{L}_j(d)\}_{j=1}^n$ defined in (5). Each sub-loss $\mathcal{L}_j(d)$ encourages the model to rank the $j$-th preferred response $y_{\tau(j)}$ above a set of less preferred responses $\{y_{\tau(k)}\}_{k=j+1}^n$, following the order $y_{\tau(j)} \succ y_{\tau(j+1)} \succ \cdots \succ y_{\tau(n)}$ for all $1 \leq j \leq n - 1$. While this recursive ranking objective explores relative preferences among dispreferred responses, it falls short in helping the LLM reject harmful dispreferred samples while suffering from high training costs.

**Inadequacy for Rejecting Harmful Responses.** Given a prompt $x$ and its ranked responses $\{y_{\tau(j)}\}_{j=1}^n$, the first response $y_{\tau(1)}$ is always the preferred and helpful output, while subsequent responses $\{y_{\tau(j)}\}_{j=2}^n$ are potentially harmful or purely dispreferred. Ideally, the training loss should

prioritize producing response $y_{\tau(1)}$ and strictly avoid generating any harmful outputs. However, the recursive nature of $\mathcal{L}_j(d)$ inadvertently encourages the model to rank potentially harmful responses $y_{\tau(j)}$ as "preferred" compared to even less preferred alternatives. This misalignment limits the model's ability to robustly reject potentially harmful content like $y_{\tau(j)}$ $(j \geq 2)$, making the PL objective insufficient for addressing tasks where the strict rejection of inappropriate outputs is paramount. The BT loss focuses only on rejecting the most dispreferred response in a pair, leaving other problematic responses unaddressed. Accordingly, the PL and BT losses inadequately address the real-world need to prohibit harmful and dispreferred responses, which is critical in many high-stakes applications as discussed earlier.

**Indiscriminate Handling of Dispreferred Responses.** Given a prompt $x$ and a set of response responses $\{y_{\tau(j)}\}_{j=1}^n$, the PL loss treats all dispreferred responses $\{y_{\tau(j)}\}_{j=2}^n$ equally as shown in the denominator in Eqn. (5) when training the model to prioritize the most preferred response without considering the inter-ranking relationship among dispreferred responses. This overlooks the varying degrees of informativeness among dispreferred responses, which could otherwise guide more effective alignment learning. The BT loss reduces rankings to pairwise comparisons, directly discarding other dispreferred responses let alone their macro-level distinctions that are crucial for capturing nuanced preferences (Sun et al., 2024; Song et al., 2024).

**Training Inefficiency.** For each prompt $x$, the PL loss $\mathcal{L}_{\text{PL}}$ requires forwarding all $n$ candidate responses $\{y_{\tau(i)}\}_{i=1}^n$ through the model to compute their rewards, followed by constructing $n$ sub-losses $\{\mathcal{L}_j(d)\}_{j=1}^n$ for back-propagation. Considering the big size of LLM, this leads to high GPU memory and computational costs, especially when dealing with long prompts or long responses. Indeed, training costs even scale linearly with the number of response candidates $n$, further severe large-scale training scenarios where computational resources and efficiency are critical considerations. While the BT loss is more efficient, its simplifications sacrifice critical preference information.

Given the limitations of the PL and BT objective in rejecting harmful responses and its high training cost, it is imperative to explore alternative strategies for alignment: robustly preventing harmful content generation while reducing training overhead. Below, we offer a more practical and effective method for aligning LLMs with real-world requirements.

### 4.2. Hard Preference Sampling for Alignment

To solve the task of interests, we propose a hard preference sampling framework (HPS). The target of the task is to train the model to reject all dispreferred and potentially harmful responses $\{y_{\tau(i)}\}_{i=2}^n$, ensuring it generates only the most preferred response $y_{\tau(1)}$ for a given prompt $x$. To this end,

for a training sample $d = (x, y_{\tau(1)}, y_{\tau(2)}, \cdots, y_{\tau(n)}) \sim \mathcal{D}$, HPS can use the training loss

$$\mathcal{L}_{\boldsymbol{\theta}} = \mathbb{E}_{d \sim \mathcal{D}} - \log\left(e^{r_{\boldsymbol{\theta}}(x, y_{\tau(1)})} / \sum_{i=1}^{n} e^{r_{\boldsymbol{\theta}}(x, y_{\tau(i)})}\right). \quad (6)$$

where the model is encouraged to rank $y_{\tau(1)}$ above all dispreferred and potentially harmful responses $\{y_{\tau(i)}\}_{i=2}^{n}$. We use the DPO implicit reward parameterization as mentioned in Eqn. (3) here. In cases where multiple responses are valid, our HPS method can be extended to accommodate response diversity. The details of this extension are provided in Appendix E.

However, this loss treats all dispreferred responses $\{y_{\tau(i)}\}_{i=2}^{n}$ equally, ignoring their varying levels of informativeness. Previous works in supervised, metric, and contrastive learning (Schroff et al., 2015; Oh Song et al., 2016; Robinson et al., 2020) demonstrate that "hard" examples — those closely resembling the correct output but still incorrect — are particularly useful for learning. In such settings, hard negatives are typically selected based on representation similarity to a positive anchor; however, in RLHF, where responses are generated autoregressively, obtaining effective sentence embeddings is impractical. Instead, in our context, hard dispreferred responses are those that are highly similar to $y_{\tau(1)}$ yet dispreferred or harmful in the reward space. Training the model to distinguish $y_{\tau(1)}$ from the hardest dispreferred response $y_{\tau(2)}$ enables it to reject less preferred responses $\{y_{\tau(i)}\}_{i=3}^{n}$ more effectively. Thus, harder dispreferred responses should be penalized more heavily during training.

**Hard Preference Sampling Framework (HPS).** Our HPS builds a distribution over the dispreferred responses as

$$q(x, y) = e^{r^*(x, y)} \cdot p(y) / Z, \quad (7)$$

where $r^*(x, y)$ is the inaccessible optimal reward model defined in Sec. 3 and can provide the ground-truth rewards, $p(y)$ is the probability distribution of the dispreferred response $y$, and $Z$ is the partition function for normalization. For each ranked response $y_{\tau(i)}$, we can either directly access its reward $r_{\text{est}}$ if available in the dataset $\mathcal{D}$ or estimate it using a pretrained human preference-aligned reward model, $r_{\text{est}}(x, y_{\tau(i)}) \approx r^*(x, y_{\tau(i)})$. Without loss of generality, we first formulate the Eqn. (6) in the expectation form:

$$\mathcal{L}_{\boldsymbol{\theta}} = \mathbb{E}_{d \sim \mathcal{D}} - \log\left(\frac{e^{r_{\boldsymbol{\theta}}(x, y_{\tau(1)})}}{e^{r_{\boldsymbol{\theta}}(x, y_{\tau(1)})} + N \cdot \mathbb{E}_{y \sim q(x, y)}\left[e^{r_{\boldsymbol{\theta}}(x, y)}\right]}\right), \quad (8)$$

where $N = n - 1$. Using the Monte Carlo importance sampling technique, Eqn. (8) becomes:

$$\mathcal{L}_{\boldsymbol{\theta}} = \mathbb{E}_{d \sim \mathcal{D}} - \log\left(\frac{e^{r_{\boldsymbol{\theta}}(x, y_{\tau(1)})}}{e^{r_{\boldsymbol{\theta}}(x, y_{\tau(1)})} + N \cdot \mathbb{E}_{y \sim p(y)}\left[e^{r_{\boldsymbol{\theta}}(x, y)} e^{r^*(x, y)} / Z\right]}\right).$$

Next, we can empirically estimate the distribution $q(x, y)$:

$$q(x, y) = e^{\gamma \cdot r_{\text{est}}(x, y)} / \sum_{i=2}^{n} e^{\gamma \cdot r_{\text{est}}(x, y_{\tau(i)})}. \quad (9)$$

Here for flexibility, we introduce a hyperparameter $\gamma > 1$ to control penalty strength in $q(x, y)$. Thus, the empirical training loss function becomes:

$$\mathcal{L}_{\boldsymbol{\theta}} = \mathbb{E}_{d \sim \mathcal{D}} - \log\left(\frac{e^{r_{\boldsymbol{\theta}}(x, y_{\tau(1)})}}{e^{r_{\boldsymbol{\theta}}(x, y_{\tau(1)})} + N \cdot \mathbb{E}_{y \sim p(y)}\left[e^{r_{\boldsymbol{\theta}}(x, y)} q(x, y)\right]}\right). \quad (10)$$

Here, harder dispreferred responses whose hardness is reflected by their bigger rewards $r_{\text{est}}(x, y)$ contribute more to the loss due to their higher weights $q(x, y)$. For instance, larger $\gamma$ sharpens the distribution, emphasizing harder dispreferred responses and enabling the model to better distinguish closely-ranked preferred and dispreferred responses.

**Reducing Training Costs with Flexible Sampling.** Although this approach improves alignment, computing rewards for all $n$ responses and backpropagating through them can be computationally expensive. To address this, we propose to sample only one dispreferred response $y \sim q(y)$ according to the importance-weighted distribution in Eqn. (9) given a prompt $x$. Thus, harder dispreferred responses will be sampled with higher probability and contribute more to the loss due to their higher $q(x, y)$. Then, we can incorporate the sampled dispreferred response $y$ into the loss function Eqn. (10) for each prompt $x$ in practice. This sampling technique works well as shown in Sec. 6 and also significantly reduces computational and memory overhead. By focusing more on the hard dispreferred responses, our method retains robust alignment while greatly improving training efficiency.

## 5. Theoretical Analysis

Here we first analyze the sample efficiency of our HPS approach and the PL method, and then theoretically justify how HPS can maximize the reward margin between the most preferred response and other hard dispreferred responses, ensuring less dispreferred or harmful generation.

### 5.1. Sample Complexity Analysis

To analyze the sample complexity of our HPS and the vanilla PL in Eqn. (5), assume $\boldsymbol{\theta}^*$ denotes the optimal human preference policy, i.e., the inaccessible reward model $r^*(x, y)$. Then given the training dataset $\mathcal{D}$ containing $m$ training samples $\{d_i\}_{i=1}^{m} = \{(x_i, \{y_{\tau_i(j)}\}_{j=1}^{n})\}_{i=1}^{m}$, define

$$\boldsymbol{\theta}_{\text{HPS}} = \arg\min_{\boldsymbol{\theta}} \mathcal{L}_{\boldsymbol{\theta}}, \quad \boldsymbol{\theta}_{\text{PL}} = \arg\min_{\boldsymbol{\theta}} \mathcal{L}_{\text{PL}}, \quad (11)$$

where $\mathcal{L}_{\boldsymbol{\theta}}$ and $\mathcal{L}_{\text{PL}}$ respectively denote our HPS loss in Eqn. (10), and the PL loss in Eqn. (5). Then we pose necessary assumptions widely used in network and RLHF analysis (Zhu et al., 2023; Li et al., 2024; Ozay, 2019).

**Assumption 1.** *a) Assume $r_\theta$ is bounded, Lipschitz and also smooth, i.e., $|r_\theta(x,y)| \leq \alpha_0, \|\nabla r_\theta(x,y)\|_2 \leq \alpha_1,$ $\|\nabla^2 r_\theta(x,y)\|_2 \leq \alpha_2$ with three constants $\alpha_0, \alpha_1$ and $\alpha_2$. b) Assume $\theta^* \in \theta_B$, where $\theta_B = \{\theta \in \mathbb{R}^d \mid \|\theta\|_2 \leq B\}$.*

Assumption 1 **a)** and **b)** pose the boundness on reward function $r_\theta$ and the optimum $\theta^*$. These boundness assumptions are often held empirically since after training network parameters are often bounded (Zhu et al., 2023).

Based on these assumptions, we can derive the following sample complexity bounds. See its proof in Appendix B.1.

**Theorem 1.** *With Assumption 1, with probability at least $1 - \delta$, the distance between the optimum solution $\theta_{HPS}$ of our HPS loss and the ground-truth optimum $\theta^*$ can be bounded:*

$$\|\theta_{HPS} - \theta^*\|_{\Sigma_\mathcal{D}} \leq \Psi_1 = C_1 \sqrt{\frac{d + \log(1/\delta)}{m\zeta^2} - \frac{16\alpha_1^2 \zeta - 4\alpha_2}{m\zeta}},$$

*where* $\zeta = \frac{1}{2 + \exp(2\alpha_0 + \ln(n-1)) + \exp(-2\alpha_0)}$ *and* $\Sigma_\mathcal{D} = \frac{2}{mn(n-1)} \sum_{i=1}^{m}\sum_{j=1}^{n}\sum_{k=j+1}^{n} z_{ijk} z_{ijk}^\top$ *in which* $z_{ijk} = \nabla r_\theta(x_i, y_{\tau_i(j)}) - \nabla r_\theta(x_i, y_{\tau_i(k)})$. *Similarly, the distance between the optimum solution $\theta_{PL}$ of PL loss and the ground-truth optimum $\theta^*$ can be bounded:*

$$\|\theta_{PL} - \theta^*\|_{\Sigma_\mathcal{D}} \leq \Psi_2 = C_2 \sqrt{\frac{n^4 e^{8\alpha_0} \cdot (d + \log(1/\delta))}{m}}.$$

Theorem 1 shows the bounded distance $\Psi_1$ between the optimum solution $\theta_{HPS}$ of our HPS loss and the ground-truth optimum $\theta^*$ and indicates its good approximation. Theorem 1 also demonstrates the distance between the optimum solution $\theta_{PL}$ of PL loss and the ground-truth $\theta^*$ is bounded by $\Psi_2$. The difference in sample complexity bounds arises from the method used to sample dispreferred responses and the aggregation process for their associated scores given a prompt. Now we compare the optimums of our HPS and PL by comparing the bounded distances $\Psi_1$ and $\Psi_2$ to the ground-truth $\theta^*$. $\Psi_1$ represents an asymptotic error bound of $\mathcal{O}\left(\frac{n}{\sqrt{m}}\right)$, while $\Psi_2$ represents an asymptotic error bound of $\mathcal{O}\left(\frac{n^2}{\sqrt{m}}\right)$. This indicates that, given the same amount of training data, our HPS achieves better preference alignment performance compared to PL. Specifically, the optimum solution $\theta_{HPS}$ derived from HPS loss is closer to the desired ground-truth $\theta^*$ than the solution obtained from PL loss. This suggests that HPS improves sample efficiency, making it advantageous in scenarios with limited data or when faster convergence to the true parameter is desired.

### 5.2. Reward Margin Analysis

For a model, we analyze its reward margin between a preferred response and the dispreferred responses under the same prompt. Intuitively, given a fixed reward for the preferred response, a larger reward margin means the lower

generation ability of the dispreferred responses, aligning with the target of human preference alignment. For this analysis, we first define the min-max loss:

$$\inf_\theta \sup_{p \in \Pi} \left\{ \mathcal{L}_\Pi = \mathop{\mathbb{E}}_{d \sim \mathcal{D}} -\log\left(\frac{e^{r_\theta(x, y_{\tau(1)})}}{e^{r_\theta(x, y_{\tau(1)})} + N \cdot \mathbb{E}_{y \sim p}[e^{r_\theta(x,y)}]}\right) \right\}$$
(12)

where $\Pi = \{p(x,\cdot) : \text{supp}(p(x,\cdot)) \subseteq \{y \in \mathcal{Y} : 1 < \tau^{-1}(y) \leq |\tau|\}\}$. Here, $\Pi$ represents a family of probability distributions whose support is restricted to elements with ranks lower than that of the sample $y_{\tau(1)}$ given the prompt $x$, where $|\tau|$ denotes the number of ranking classes.

**Theorem 2.** *Let $\mathcal{L}_\Pi^* = \sup_{p \in \Pi} \mathcal{L}_\Pi$. Then it holds the convergence: $\mathcal{L}_\theta \to \mathcal{L}_\Pi^*$ as $\gamma \to \infty$ where $\mathcal{L}_\theta$ is our HPS loss.*

Theorem 2 establishes that when $\gamma \to \infty$, then our HPS training loss $\mathcal{L}_\theta$ in (6) would converge to $\mathcal{L}_\Pi^*$ which is the loss under the hardest dispreferred distribution. Since $\mathcal{L}_\Pi^*$ samples the hardest dispreferred responses, optimizing $\mathcal{L}_\Pi^*$ encourages the model to identify the preferred response and hardest dispreferred responses, which is the desired training. This is because as discussed before, the works in supervised, metric, and contrastive learning (Schroff et al., 2015; Oh Song et al., 2016; Robinson et al., 2020) demonstrate that "hard" examples—those closely resembling the correct output but still incorrect—are particularly useful for learning. In our context, training the model to distinguish the preferred response from the hardest dispreferred response enables it to reject less preferred responses more effectively.

Furthermore, to examine the global minimizer of our HPS training loss $\mathcal{L}_\theta$, we analyze the optima of the training loss $\mathcal{L}_\Pi^*$ in Eqn. (12), since we have proved their good approximation in Theorem 2. Without loss of generality, when the number of dispreferred response samples $N = n - 1 \to \infty$, we can remove the $\log N$ from the HPS training loss $\mathcal{L}_\theta$ as it does not change the minimizers and the geometry of the loss surface, and obtain a limiting objective:

$$\mathcal{L}_\theta^\infty = \mathop{\mathbb{E}}_{d \sim \mathcal{D}} \left[ -\log\left(\frac{e^{r_\theta(x, y_{\tau(1)})}}{\mathbb{E}_{y \sim p}[e^{r_\theta(x,y)}]}\right) \right].$$
(13)

Now we are ready to give our results in Theorem 3 whose proof is in Appendix B.2.2.

**Theorem 3.** *Assume the ranking set $\tau$ is a finite set. Let $\mathcal{L}_\theta^{\infty,*} = \sup_{p \in \Pi} \mathcal{L}_\theta^\infty$ and $\theta^* = \arg\min_\theta \mathcal{L}_\theta^{\infty,*}$. Then $\theta^*$ is also the solution to the following problem:*

$$\theta^* = \arg\max_\theta \left(r_\theta(x, y_{\tau(1)}) - \max_{1 < j \leq |\tau|} r_\theta(x, y_{\tau(j)})\right).$$
(14)

Theorem 3 implies that the minimizer $\theta^* = \arg\min_\theta \mathcal{L}_\theta^{\infty,}$ is equivalent to the one that maximizes the margin between the reward of the most preferred response, represented by $r_\theta(x, y_{\tau(1)})$, and the reward of the hardest dispreferred responses, represented by $\max_{1 < j \leq |\tau|} r_\theta(x, y_{\tau(j)})$. So the optimum $\theta^*$ of our HPS loss aims to maximize the reward

*Table 1.* Reward margin definitions of $\mathbf{RM}_{\text{DPO}}$ and $\mathbf{RM}_{\text{R-DPO}}$ induced by DPO (Rafailov et al., 2024) and R-DPO (Park et al., 2024). For a sample $(x, y_{\tau(1)}, y_{\tau(2)})$, they denote the margin of implicit rewards between the preferred $y_{\tau(1)}$ and dispreferred $y_{\tau(2)}$, where $|y_{\tau(1)}|$ and $|y_{\tau(2)}|$ are the respective response lengths.

| Type | Method | Reward Margin Formula |
|------|--------|----------------------|
| $\mathbf{RM}_{\text{DPO}}$ | BT-DPO | $\mathbf{RM}_{\text{DPO}} = \log \frac{\pi_{\boldsymbol{\theta}}(y_{\tau(1)}|x)}{\pi_{\text{ref}}(y_{\tau(1)}|x)} - \log \frac{\pi_{\boldsymbol{\theta}}(y_{\tau(2)}|x)}{\pi_{\text{ref}}(y_{\tau(2)}|x)}$ |
| $\mathbf{RM}_{\text{R-DPO}}$ | R-DPO | $\mathbf{RM}_{\text{R-DPO}} = \mathbf{RM}_{\text{DPO}} - 0.01(|y_{\tau(1)}| - |y_{\tau(2)}|)$ |

margin between the preferred response and its closest dispreferred response. This guarantees that the model learns a robust distinction between preferred and dispreferred responses, and enjoys a better alignment performance with much less dispreferred or harmful generation.

# 6. Experiments

**Baselines.** In our experiments, we employ a supervised fine-tuned Llama3-8B checkpoint `RLHFlow/Llama3-SFT-v2.0` (Dong et al., 2024; Dubey et al., 2024) as both the naive baseline SFT and the reference model. In addition, we integrate three preference modeling strategies — BT, PL, and our proposed HPS — into several implicit reward parameterization frameworks, including DPO (Rafailov et al., 2024), EXO (Ji et al., 2024a), IPO (Azar et al., 2024), SPPO (Wu et al., 2024), and NCA (Chen et al., 2024). For example, DPO-PL refers to the configuration where Llama3-8B is fine-tuned using the DPO implicit reward parameterization under a PL preference model.

**Datasets.** We use two popular datasets, HH-RLHF (Bai et al., 2022a) and PKU-SafeRLHF (Ji et al., 2024b), focusing on helpfulness and safety (Lambert et al., 2024; Fourrier et al., 2024). HH-RLHF is multi-turn, while PKU-SafeRLHF contains single question-answer pairs. Each prompt in the datasets includes two responses with human-rated preferences. Following prior work (Song et al., 2024), we expand response data by generating 100 responses using `RLHFlow/Llama3-v2-DPO` (Dong et al., 2024) per prompt. The corresponding rewards are computed via `Skywork/Skywork-Reward-Llama3`, a safety-aligned reward model (Liu et al., 2024a) ranked among the top 10 on the RewardBench (Lambert et al., 2024).

**Evaluation Metrics.** We evaluate response quality and harmful content rejection. We use BLEU (Papineni et al., 2002) to assess the text quality by comparing responses to ground-truth preferred answers. To evaluate alignment with human preference, we also adopt a powerful reward model `RLHFlow/ArmoRM-Llama3` (Wang et al., 2024), which is different from the one used during training, to measure the level of human preference gained. Importantly, we compute

reward margins (RMs) from various implicit reward models (Table 1) to quantify the gap between preferred and dispreferred responses, where higher RM scores indicate better preference alignment without harmful or biased outputs.

Human evaluation remains the gold standard for assessing response quality, where annotators compare two responses per question to select the better one or declare a tie. Thus, we conduct a user study to evaluate the performance of DPO and its variants. Moreover, recent LLMs like Qwen 2.5 (Yang et al., 2024) closely align with human preferences, validated by benchmarks like the Open LLM Leaderboard (Fourrier et al., 2024), chatbot-arena-leaderboard (Chiang et al., 2024), and RewardBench (Lambert et al., 2024). We use Qwen2.5-72B-Instruct to assess response quality and win rates. For evaluation robustness, we sample $N = 5$ responses per method and report the highest-scoring one for each metric.

**Implementation.** Due to computational constraints, we apply LoRA (Hu et al., 2021) for efficient fine-tuning with a rank of 8 and scaling factor $\alpha = 16$. The KL penalty strength $\beta$ is set to 0.1, following DPO. The ablation study about sensitivity of $\beta$ in DPO can be found in Appendix C. We fine-tune all methods for 2 epochs use AdamW optimizer (Loshchilov, 2017) with a learning rate of $5.0 \times 10^{-7}$ over 2 and a cosine learning rate scheduler. We set the sequence length as $2,048$ tokens for both training and inference, with a sampling temperature of 0.9 during inference. In our HPS setting, we use the annotated scalar reward as the estimated reward $r_{\text{est}}$ for each response in Eqn. (9). The scaling factor $\gamma$ is scheduled to linearly increase from -5 to 5, with updates applied at every 20% interval of the training process. The GPU fine-tuning time for the PL-based, BT-based, and HPS-based methods is $168.4 \pm 1.9$ hours, $62.8 \pm 1.1$ hours, and $64.4 \pm 0.8$ hours, respectively. This demonstrates that our proposed HPS method can significantly improve efficiency compared to the PL-based method, achieving a 61.76% reduction in fine-tuning time, and validates Theorem 1. More implementation details can be found in Appendix D.

## 6.1. Fine-Tuning Setting

We integrate HPS into various alignment approaches and fine-tune LLMs on the HH-RLHF and PKU-SafeRLHF datasets. Table 2 reports BLEU, Reward, and Reward Margin, revealing two key findings: **1)** HPS achieves comparable performance on BLEU and Reward metrics. For instance, on HH-RLHF, HPS-based methods achieve BLEU scores around 0.231, similar to BT-based methods. This shows that HPS does not affect the quality of the generation of preferred content. **2)** HPS significantly improves Reward Margin, reducing harmful or unhelpful responses. Traditional methods like DPO-PL and SPPO exhibit nega-

Table 2. Result comparison under fine-tuning setting. See Reward Margins in Table 1.

| Method | HH-RLHF | | | | PKU-SafeRLHF | | | |
|---|---|---|---|---|---|---|---|---|
| | BLEU↑ | Reward↑ | $RM_{DPO}$↑ | $RM_{R\text{-}DPO}$↑ | BLEU↑ | Reward↑ | $RM_{DPO}$↑ | $RM_{R\text{-}DPO}$↑ |
| SFT Model | 0.220 | 0.425 | - | - | 0.294 | 0.406 | - | - |
| DPO-PL | 0.230 | 0.430 | -0.795 | -1.448 | 0.305 | 0.412 | -6.852 | -5.961 |
| DPO-BT | 0.230 | **0.431** | 0.349 | -0.455 | **0.306** | **0.417** | -5.441 | -6.167 |
| DPO-HPS | **0.232** | 0.430 | **2.723** | **2.040** | **0.306** | 0.407 | **-5.359** | **-5.851** |
| EXO-PL | **0.232** | **0.432** | -0.724 | -1.406 | 0.303 | 0.409 | **-5.455** | -6.128 |
| EXO-BT | 0.231 | 0.430 | 0.816 | 0.215 | **0.324** | 0.421 | -5.553 | -6.164 |
| EXO-HPS | **0.232** | **0.432** | **1.079** | **0.410** | 0.314 | **0.425** | -5.495 | **-6.031** |
| IPO-PL | 0.223 | **0.429** | -5.199 | -5.264 | 0.309 | 0.401 | -66.337 | -66.946 |
| IPO-BT | **0.232** | 0.428 | -0.382 | -1.254 | **0.310** | 0.405 | -23.070 | -23.678 |
| IPO-HPS | 0.231 | 0.424 | **-0.321** | **-0.926** | 0.308 | **0.406** | **-21.607** | **-22.215** |
| SPPO-PL | 0.225 | 0.430 | -7.630 | -7.674 | 0.297 | 0.413 | -67.474 | -68.082 |
| SPPO-BT | **0.231** | **0.432** | -0.978 | -1.411 | 0.297 | 0.433 | -13.442 | -14.050 |
| SPPO-HPS | **0.231** | **0.432** | **-0.969** | **-1.302** | **0.298** | **0.435** | **-5.273** | **-5.881** |
| NCA-PL | 0.221 | 0.431 | -5.760 | -5.819 | 0.300 | **0.411** | -70.910 | -71.518 |
| NCA-BT | 0.229 | **0.432** | -1.702 | -3.121 | **0.305** | 0.410 | -5.135 | -5.644 |
| NCA-HPS | **0.231** | **0.432** | **-0.822** | **-1.268** | 0.304 | **0.411** | **-5.109** | **-5.318** |

Table 3. Human evaluation comparing **SFT**, **DPO-BT**, **DPO-PL**, and **DPO-HPS** on user study dataset under fine-tuning conditions.

| Method | Quality Score |
|---|---|
| SFT | 3.63 |
| DPO-BT | 3.82 |
| DPO-PL | 3.69 |
| DPO-HPS | **3.93** |

Table 4. Win rates (%) of **DPO-HPS** compared to the baselines **SFT**, **DPO-BT**, and **DPO-PL** under fine-tuning conditions.

| Dataset | Metric | SFT | DPO-BT | DPO-PL |
|---|---|---|---|---|
| **HH-RLHF** | Win | 63.875 | 58.712 | 57.612 |
| | Lose | 13.225 | 13.600 | 13.875 |
| | Tie | 22.900 | 27.687 | 28.513 |
| **PKU-Safety** | Win | 67.100 | 56.100 | 57.650 |
| | Lose | 5.150 | 11.150 | 10.350 |
| | Tie | 27.750 | 32.750 | 32.000 |

tive $RM_{DPO}$ and $RM_{R\text{-}DPO}$ values, indicating a higher likelihood of harmful outputs (e.g., DPO-PL: $RM_{DPO}$ of $-0.795$, $RM_{R\text{-}DPO}$ of $-1.448$). In contrast, DPO-HPS shows $RM_{DPO}$ of 2.723 and $RM_{R\text{-}DPO}$ of 2.040, reflecting improvements of $442.51\%$ and $240.88\%$. This validates Theorem 3, confirming HPS leads to stronger rejection of harmful responses.

**User Study Evaluation.** For human evaluation, we created the user study dataset by selecting 15 prompt questions from the HH-RLHF test dataset and 15 prompt questions from the PKU-Safety test dataset. Then, for each question, four responses — generated by SFT, DPO-BT, DPO-PL, and DPO-HPS — are evaluated by 20 different human raters. Each rater assigns an overall quality score to each response on the Likert scale of 1-5. To eliminate bias, the models are anonymized, and the order of responses is randomized for each task. Details of the evaluation methodology are provided in Appendix D.

As shown in Table 3, DPO-HPS achieves the highest quality score (3.93), outperforming all other methods, including SFT, DPO-BT, and DPO-PL, thereby demonstrating its effectiveness in enhancing response helpfulness under the fine-tuning setting.

**Win Rate Evaluation.** Unlike reward models that may distort human preferences, recent advances in instruction-tuned LLMs offer a scalable and reliable alternative for evaluating human preferences. Thus, we use Qwen-2.5-Instruct to assess response quality on the Likert scale of 0-5 and win rates, where Qwen 2.5 closely aligns with human preferences, validated by benchmarks like the Open LLM Leaderboard (Fourrier et al., 2024), chatbot-arena-leaderboard (Chiang et al., 2024), and RewardBench (Lambert et al., 2024). Details of the evaluation methodology are provided in Appendix D.

As shown in Table 4, DPO-HPS consistently outperforms the baselines — SFT, DPO-BT, and DPO-PL — across both the HH-RLHF and PKU-Safety datasets, achieving an impressive win rate of approximately 60%. This result underscores HPS's superior alignment with human preferences and is consistent with the reward model evaluation.

### 6.2. Transfer Learning Setting

After fine-tuning LLMs on HH-RLHF (PKU-SafeRLHF), we evaluate their transferability on PKU-SafeRLHF (HH-RLHF) to assess generation quality and harmfulness rejection in a transfer learning setting.

Table 5 presents the results, leading to two key conclusions. First, HPS achieves comparable BLEU and Reward scores, demonstrating strong transferability. Despite dataset differences, HPS-based methods perform on par with baselines.

*Table 5.* Result comparison under transfer learning setting. See Reward Margins in Table 1.

| Method | HH-RLHF | | | | PKU-SafeRLHF | | | |
|---|---|---|---|---|---|---|---|---|
| | BLEU↑ | Reward↑ | $RM_{DPO}$↑ | $RM_{R\text{-}DPO}$↑ | BLEU↑ | Reward↑ | $RM_{DPO}$↑ | $RM_{R\text{-}DPO}$↑ |
| **DPO-PL** | **0.219** | 0.435 | -5.122 | -5.638 | 0.307 | **0.407** | 0.144 | -0.464 |
| **DPO-BT** | **0.219** | 0.437 | -5.053 | -5.736 | 0.308 | **0.407** | 1.346 | 1.738 |
| **DPO-HPS** | **0.219** | **0.438** | **-4.816** | **-5.499** | 0.310 | **0.407** | **5.725** | **5.116** |
| **EXO-PL** | **0.222** | 0.436 | -5.183 | -6.166 | 0.303 | **0.409** | 0.845 | 0.237 |
| **EXO-BT** | 0.203 | 0.442 | -4.825 | -5.508 | 0.306 | 0.407 | 2.710 | 2.102 |
| **EXO-HPS** | 0.197 | **0.444** | **-4.583** | **-5.266** | 0.310 | 0.407 | **2.903** | **3.495** |
| **IPO-PL** | **0.189** | 0.439 | -60.129 | -60.809 | **0.306** | 0.406 | 2.294 | 1.686 |
| **IPO-BT** | **0.189** | 0.446 | -21.821 | -22.504 | **0.306** | 0.405 | 4.393 | 3.786 |
| **IPO-HPS** | 0.187 | **0.449** | **-20.938** | **-21.255** | 0.305 | **0.409** | **5.386** | **4.779** |
| **SPPO-PL** | 0.222 | 0.441 | -59.290 | -59.605 | **0.309** | 0.406 | -8.160 | -8.197 |
| **SPPO-BT** | 0.181 | **0.449** | -11.848 | -13.030 | **0.309** | **0.407** | -0.134 | -0.843 |
| **SPPO-HPS** | **0.275** | 0.435 | **-4.184** | **-6.117** | **0.309** | **0.407** | **-0.101** | **-0.810** |
| **NCA-PL** | 0.214 | 0.433 | -61.084 | -61.762 | 0.306 | **0.409** | -8.230 | -8.267 |
| **NCA-BT** | **0.226** | 0.435 | **-4.673** | **-5.557** | **0.309** | 0.407 | -0.373 | -0.982 |
| **NCA-HPS** | 0.224 | **0.436** | -5.378 | -5.562 | 0.308 | 0.407 | **-0.102** | **-0.711** |

*Table 6.* Ablation results with response number under fine-tuning setting. See Reward Margins in Table 1.

| Number | Method | BLEU↑ | Reward↑ | $RM_{DPO}$↑ | $RM_{R\text{-}DPO}$↑ |
|---|---|---|---|---|---|
| 5 | **DPO-BT** | **0.229** | **0.432** | 0.166 | -0.516 |
| | **DPO-HPS** | **0.229** | 0.431 | **0.600** | **-0.273** |
| 20 | **DPO-BT** | **0.231** | 0.430 | 0.227 | -0.490 |
| | **DPO-HPS** | 0.224 | **0.432** | **0.822** | **-0.181** |
| 50 | **DPO-BT** | **0.230** | **0.431** | 0.279 | -0.507 |
| | **DPO-HPS** | **0.230** | **0.431** | **1.645** | **1.037** |
| 100 | **DPO-BT** | 0.230 | **0.431** | 0.349 | -0.455 |
| | **DPO-HPS** | **0.232** | 0.430 | **2.723** | **2.040** |

For example, DPO-HPS achieves BLEU scores of $0.219$ (HH-RLHF) and $0.310$ (PKU-Safety), similar to DPO-BT ($0.219$ and $0.308$). This consistency suggests that HPS effectively transfers learned preferences and linguistic structures.

Moreover, HPS improves harmfulness rejection robustness, as reflected in Reward Margin. HPS consistently outperforms baselines in terms of $RM_{DPO}$ and $RM_{R\text{-}DPO}$, showing better generalization of safety properties. Notably, DPO-HPS achieves $RM_{DPO}$ of $5.725$ on PKU-Safety, compared to DPO-BT's $RM_{DPO}$ of $1.346$. Additionally, HPS excels in transfer tasks, with EXP-HPS achieving $RM_{DPO}$ of $2.903$ on PKU-Safety, significantly surpassing its fine-tuned counterparts which has $RM_{DPO}$ of $-5.495$, demonstrating its potential for safer and more effective cross-domain transfer.

### 6.3. Ablation Study

We examine the impact of the total number of responses on preference optimization performance during fine-tuning, using $5, 20, 50,$ and $100$ responses per prompt. From Table 6, one can observe that while BLEU and Reward scores remain stable across response sizes for both DPO-BT and DPO-HPS, notable differences appear in $RM_{DPO}$ and $RM_{R\text{-}DPO}$. As response size increases, $RM_{R\text{-}DPO}$ shows a pronounced improvement, particularly for DPO-HPS, which achieves

a remarkable $RM_{R\text{-}DPO}$ of $2.040$ at $100$ responses, far surpassing DPO-BT's $-0.455$. This suggests that DPO-HPS benefits more from larger response sets, enhancing preference alignment. Additionally, the consistent increase in $RM_{DPO}$ for DPO-HPS suggests a cumulative learning effect, indicating that DPO-HPS scales better and achieves superior preference optimization with larger response sizes. The ablation study under the transfer learning setting is provided in Appendix C.

## 7. Conclusion

Ensuring LLMs align with human preferences is crucial for building safe and controllable AI systems. We introduce Hard Preference Sampling (HPS), a novel framework that improves preference alignment by prioritizing the most preferred responses while effectively rejecting harmful and dispreferred ones. HPS enhances rejection capabilities by emphasizing "hard" dispreferred responses and employs a single-sample Monte Carlo strategy to reduce computational costs. Theoretically, it improves sample efficiency and maximizes reward margins, ensuring clearer distinctions between preferred and dispreferred responses. Experiments on HH-RLHF and PKU-Safety datasets demonstrate HPS's effectiveness, achieving strong BLEU and reward scores while significantly reducing harmful content generation.

**Limitations.** Due to budget constraints, our experiments rely on open-source LLMs to estimate the win rate. More powerful instruct LLMs, such as GPT-4 (Achiam et al., 2023) and Claude 3 (Anthropic, 2023), may offer more accurate and robust evaluations and will be considered when additional resources become available.

## Acknowledgements

This research is supported by the Ministry of Education, Singapore, under its AcRF Tier 2 Funding (Proposal ID: T2EP20224-0048) and its AcRF Tier 1 grant (project ID: 23-SIS-SMU-070 and 23-SIS-SMU-063). Any opinions, findings and conclusions or recommendations expressed in this material are those of the author(s) and do not reflect the views of the Ministry of Education, Singapore.

## Impact Statement

The data used in this research may include sensitive or potentially offensive content, intended solely for academic and scientific purposes. The opinions expressed within this data do not represent the views of the authors. We remain committed to fostering the development of AI technologies which align with ethical standards and reflect societal values.

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

# A. Reinforcement Learning from Human Feedback

## A.1. Bradley-Terry model

Then sampling samples from $p_{\text{BT}}^*$, one can construct a dataset $\mathcal{D} = \{(x_i, y_{\tau_i(1)}, y_{\tau_i(2)})\}_{i=1}^m$, where each instance consists of one prompt $x_i$ and 2 responses $y_{\tau_i(1)}, y_{\tau_i(2)}$ followed the user-specified ranking. Using this dataset, we can train a reward model $r_{\boldsymbol{\theta}}$ parameterized by $\boldsymbol{\theta}$ by approaching the task as a classification problem. Specifically, we frame the training objective as minimizing the negative log-likelihood loss:

$$
\begin{aligned}
\mathcal{L}_{\text{BT}} &= - \mathop{\mathbb{E}}_{(x_i, y_{\tau_i(1)}, y_{\tau_i(2)}) \sim \mathcal{D}} \left[ \log \sigma \left( r_{\boldsymbol{\theta}}(x_i, y_{\tau_i(1)}) - r_{\boldsymbol{\theta}}(x_i, y_{\tau_i(2)}) \right) \right] \\
&= - \sum_{i=1}^m \log \sigma \left( r_{\boldsymbol{\theta}}(x_i, y_{\tau_i(1)}) - r_{\boldsymbol{\theta}}(x_i, y_{\tau_i(2)}) \right).
\end{aligned}
\tag{15}
$$

## A.2. Listwise Preference Optimization

### A.2.1. SLiC-HF

SLiC-HF integrates the sequence-level contrastive method SLiC with human preference rankings:

$$
\mathcal{L}(\theta) = \max(0, \delta - \log(\pi_\theta(y^+|x)) + \log(\pi_\theta(y^-|x)) - \lambda \log(\pi_\theta(y_{ref}|x))).
\tag{16}
$$

$y^+$, $y^-$, and $y_{ref}$ denote the positive, negative, and reference sequences, respectively. $\delta$ is a margin hyperparameter and $\lambda$ is a regularization weight.

### A.2.2. LiPO-$\lambda$

LiPO-$\lambda$ employs a listwise ranking objective with a Lambda weight $\Delta_{i,j}$. Given a list of responses $\boldsymbol{y} = (y_1, \ldots, y_K)$,

$$
\mathcal{L}_{LiPO} = \mathbb{E}_{(x, \boldsymbol{y}, \psi) \sim \mathcal{D}} \left[ \sum_{\psi_i > \psi_j} \Delta_{i,j} \log(1 + e^{-(s_i - s_j)}) \right],
\tag{17}
$$

where $\Delta_{i,j} = |2^{\psi_i} - 2^{\psi_j}| \cdot |\frac{1}{\log(1+\tau(i))} - \frac{1}{\log(1+\tau(j))}|$. Here, $\psi_i$ is the true reward score of response $y_i$, and $s_i = \beta \log \frac{\pi_\theta(y_i|x)}{\pi_{ref}(y_i|x)}$ is the implicit DPO reward. The rank position of $y_i$ in the ordering induced by $\mathbf{s} = (s_1, \ldots, s_K)$ is denoted as $\tau(i)$. The Lambda weight assigns greater importance to response pairs with larger preference gaps, i.e., $\psi_i - \psi_j$.

## A.3. Reward Modelling

### A.3.1. KTO

KTO (Ethayarajh et al., 2024) defines a type of reward to construct human-aware losses (HALOs), which is in the form

$$
r_{\text{KTO}}(x, y) = l(y) \log \frac{\pi_{\boldsymbol{\theta}}(y|x)}{\pi_{\text{ref}}(y|x)},
\tag{18}
$$

where $\boldsymbol{\theta}$ denotes the trainable parameters of the model $\pi_{\boldsymbol{\theta}}$ being aligned, $\pi_{\text{ref}}$ is the reference model, and $l : \mathcal{Y} \to \mathbb{R}^+$ is a normalizing factor.

### A.3.2. SimPO

SimPO (Meng et al., 2024) identifies the discrepancy between DPO's reward and the likelihood metric used for generation, and proposes an alternative reference-free reward training loss:

$$
r_{\text{SimPO}}(x, y) = \frac{\beta}{|y|} \log \pi_{\boldsymbol{\theta}}(y|x) = \frac{\beta}{|y|} \sum_{i=1}^{|y|} \log \pi_{\boldsymbol{\theta}}(y_i|x, y_{<i}),
\tag{19}
$$

where $|y|$ is the length of the response $y$, and $y_{<i}$ is the set of tokens in the sentence $y$ before the token $y_i$.

## B. Theoretical Analysis

### B.1. Sample Efficiency Analysis

**Theorem.** *Let $\mathcal{D}$ be a given dataset. Under certain regularity conditions, the maximum likelihood estimators $\hat{\boldsymbol{\theta}}_{HDR}$ and $\hat{\boldsymbol{\theta}}_{PL}$, corresponding to the the hard sampling loss $\mathcal{L}_{\boldsymbol{\theta}}$ and PL loss $\mathcal{L}_{PL}$, respectively, satisfies the following with probability at least $1 - \delta$:*

$$\|\boldsymbol{\theta}_{HPS} - \boldsymbol{\theta}^*\|_{\Sigma_{\mathcal{D}}} \leq C_1 \cdot \sqrt{\frac{d + \log\left(\frac{1}{\delta}\right)}{m\zeta^2(N)} - \frac{16\alpha_1^2\zeta(N) - 4\alpha_2}{m \cdot \zeta(N)}} = \mathcal{O}\left(\frac{n}{\sqrt{m}}\right)$$

*and*

$$\|\boldsymbol{\theta}_{PL} - \boldsymbol{\theta}^*\|_{\Sigma_{\mathcal{D}}} \leq C_2 \cdot \sqrt{\frac{n^4 e^{8\alpha_0} \cdot \left(d + \log\left(\frac{1}{\delta}\right)\right)}{m}} = \mathcal{O}\left(\frac{n^2}{\sqrt{m}}\right),$$

*where*

$$\Sigma_{\mathcal{D}} = \frac{2}{mn(n-1)} \sum_{i=1}^{m} \sum_{j=1}^{n} \sum_{k=j+1}^{n} \left(\nabla r_{\boldsymbol{\theta}}\left(x_i, y_{\tau_i(j)}\right) - \nabla r_{\boldsymbol{\theta}}\left(x_i, y_{\tau_i(k)}\right)\right)\left(\nabla r_{\boldsymbol{\theta}}\left(x_i, y_{\tau_i(j)}\right) - \nabla r_{\boldsymbol{\theta}}\left(x_i, y_{\tau_i(k)}\right)\right)^T$$

*and*

$$\zeta(N) = \frac{1}{2 + \exp(2\alpha_0 + \ln(N)) + \exp(-2\alpha_0)}.$$

*Therefore, the error bound of $\hat{\boldsymbol{\theta}}_{HDR}$ is tighter than that of $\boldsymbol{\theta}_{PL}$, i.e., $\|\hat{\boldsymbol{\theta}}_{HDR} - \boldsymbol{\theta}^*\|_{\Sigma_{\mathcal{D}}} \leq \|\boldsymbol{\theta}_{PL} - \boldsymbol{\theta}^*\|_{\Sigma_{\mathcal{D}}}$. In other words, $\hat{\boldsymbol{\theta}}_{HDR}$ is more efficient than $\hat{\boldsymbol{\theta}}_{PL}$.*

*Proof.* We begin by proving Theorem 1.

**Analysis on $\mathcal{L}_{\boldsymbol{\theta}}$**  We first analyze the asymptotic efficiency and estimation error of estimator induced by $\mathcal{L}_{\boldsymbol{\theta}}$. We consider the general RLHF setting in a dataset $\mathcal{D}$ with $m$ sample:

$$\mathcal{L}_{\boldsymbol{\theta}} = -\frac{1}{m} \sum_{i=1}^{m} \log\left(\frac{e^{r_{\boldsymbol{\theta}}(x_i, y_{\tau_i(1)})}}{e^{r_{\boldsymbol{\theta}}(x_i, y_{\tau_i(1)})} + N \cdot \mathbb{E}_{y_i \sim q(x,y)}\left[e^{r_{\boldsymbol{\theta}}(x, y_i)}\right]}\right)$$

The maximum likelihood estimator (MLE) $\boldsymbol{\theta}_{\text{HPS}}$ aims at minimizing the negative log likelihood, defined as:

$$\boldsymbol{\theta}_{\text{HPS}} \in \arg\min_{\boldsymbol{\theta} \in \boldsymbol{\theta}_B} \mathcal{L}_{\boldsymbol{\theta}}.$$

When the minimizer is not unique, we take any of the $\boldsymbol{\theta}_{\text{HPS}}$ achieve the minimum.

To simplify the notation, for a fixed sampling method $q$, we let $g_{\boldsymbol{\theta}}^i = r_{\boldsymbol{\theta}}\left(x_i, y_{\tau_i(1)}\right) - \ln(N) - \mathbb{E}_{y_i \sim q}[r_{\boldsymbol{\theta}}(x, y_i)]$. We can see that the gradient of $\mathcal{L}_{\boldsymbol{\theta}}$ takes the form:

$$\nabla \mathcal{L}_{\boldsymbol{\theta}}(\boldsymbol{\theta}) = -\frac{1}{m} \sum_{i=1}^{m} \log\left[\mathbf{1}\left[\tau^{-1}(x_i, y_{\tau_i(1)}) = 1\right] \frac{\exp\left(-g_{\boldsymbol{\theta}}^i\right)}{1 + \exp\left(-g_{\boldsymbol{\theta}}^i\right)} - \mathbf{1}\left[\tau^{-1}(x_i, y_{\tau_i(1)}) \neq 1\right] \frac{1}{1 + \exp\left(-g_{\boldsymbol{\theta}}^i\right)}\right] \nabla g_{\boldsymbol{\theta}}^i.$$

And the Hessian of $\mathcal{L}_{\boldsymbol{\theta}}$ is

$$\nabla^2 \mathcal{L}_{\boldsymbol{\theta}}(\boldsymbol{\theta}) = \frac{1}{m} \sum_{i=1}^{m} \left(\frac{\exp\left(g_{\boldsymbol{\theta}}^i\right)}{\left(1 + \exp\left(g_{\boldsymbol{\theta}}^i\right)\right)^2} \cdot \nabla g_{\boldsymbol{\theta}}^i \nabla g_{\boldsymbol{\theta}}^{i\,\mathrm{T}} - \frac{\mathbf{1}\left[\tau^{-1}(x_i, y_{\tau_i(1)}) = 1\right] \cdot \exp\left(-g_{\boldsymbol{\theta}}^i\right)}{1 + \exp\left(-g_{\boldsymbol{\theta}}^i\right)} \cdot \nabla^2 g_{\boldsymbol{\theta}}^i\right.$$
$$\left. + \frac{\mathbf{1}\left[\tau^{-1}(x_i, y_{\tau_i(1)}) \neq 1\right] \cdot \exp\left(g_{\boldsymbol{\theta}}^i\right)}{1 + \exp\left(g_{\boldsymbol{\theta}}^i\right)} \cdot \nabla^2 g_{\boldsymbol{\theta}}^i\right)$$

We bound $\mathbb{E}_{y_i \sim q}[r_{\boldsymbol{\theta}}(x, y_i)]$ using the assumption: $-\alpha_0 \leq \mathbb{E}_{y_i \sim q}[r_{\boldsymbol{\theta}}(x, y_i)] \leq \alpha_0$.

Thus,

$$\frac{\exp\left(g_{\boldsymbol{\theta}}^i\right)}{\left(1 + \exp\left(g_{\boldsymbol{\theta}}^i\right)\right)^2} \geq \zeta\left(N\right),$$

where $\zeta\left(N\right) = \frac{1}{2 + \exp(2\alpha_0 + \ln(N)) + \exp(-2\alpha_0)}$.

We say $\Sigma \succeq \Sigma'$ if $\Sigma - \Sigma'$ is positive semidefinite. Based on Assumption 1, we have

$$\nabla^2 \mathcal{L}_{\boldsymbol{\theta}}(\boldsymbol{\theta}) \succeq \frac{1}{m} \sum_{i=1}^m \left[ \zeta\left(N\right) \nabla g_{\boldsymbol{\theta}}^i \nabla g_{\boldsymbol{\theta}}^{i\,\mathrm{T}} - 2\alpha_2 I \right]. \tag{20}$$

Based on the Lipschitz gradient assumption, we also know that $\|\nabla g_{\boldsymbol{\theta}}^i - \nabla g_{\boldsymbol{\theta}^*}^i\|_2 \leq 4\alpha_1$. Let $u = \nabla g_{\boldsymbol{\theta}}^i - \nabla g_{\boldsymbol{\theta}^*}^i$, we have:

$$\nabla^2 \mathcal{L}_{\boldsymbol{\theta}}(\boldsymbol{\theta}) \succeq \frac{1}{m} \sum_{i=1}^m \zeta\left(N\right) \left(\nabla g_{\boldsymbol{\theta}^*}^i + u\right) \left(\nabla g_{\boldsymbol{\theta}^*}^{i\,\mathrm{T}} + u\right) - 2\alpha_2 I$$

$$\succeq \frac{1}{m} \sum_{i=1}^m \zeta\left(N\right) \nabla g_{\boldsymbol{\theta}^*}^i \nabla g_{\boldsymbol{\theta}^*}^{i\,T} + \zeta\left(N\right) \left(\nabla g_{\boldsymbol{\theta}^*}^i u^T + u \nabla g_{\boldsymbol{\theta}^*}^{i\,T}\right) - 2\alpha_2 I$$

Using the Cauchy's Inequality, for arbitrary $v \in \mathbb{R}^d$, $u^T v \leq \|u\|_2 \|v\|_2 \leq 4\alpha_1 \|v\|_2$, $v^T \nabla g_{\boldsymbol{\theta}^*}^i \leq \alpha_1 \|v\|_2$, where $\|x\|_2 = \sqrt{\sum_{i=1}^n x^{(i)^2}}$, this gives that:

$$v^T \nabla^2 \mathcal{L}_{\boldsymbol{\theta}}(\boldsymbol{\theta}) v \geq \frac{\zeta\left(N\right)}{m} \|Xv\|_2^2 + \frac{8\alpha_1^2 \zeta(N) - 2\alpha_2}{m} \|v\|_2^2,$$

where $X \in \mathbb{R}^{m \times d}$ has the vector $\nabla g_{\boldsymbol{\theta}^*}^i \in \mathbb{R}^d$ as its $i^{th}$ row.

Thus, if we introduce the error vector $\Delta := \boldsymbol{\theta}_{\mathrm{HPS}} - \boldsymbol{\theta}^*$, then we may conclude that:

$$\mathcal{L}_{\boldsymbol{\theta}}(\boldsymbol{\theta}^* + \Delta) - \mathcal{L}_{\boldsymbol{\theta}}(\boldsymbol{\theta}^*) - \langle \nabla \mathcal{L}_{\boldsymbol{\theta}}(\boldsymbol{\theta}^*), \Delta \rangle$$

$$\geq \frac{\zeta\left(N\right)}{m} \|X\Delta\|_2^2 + \frac{8\alpha_1^2 \zeta(N) - 2\alpha_2}{m} \|\Delta\|_2^2$$

$$\geq \zeta\left(N\right) \|\Delta\|_{\Sigma_{\mathcal{D}}}^2 + \frac{8\alpha_1^2 \zeta(N) - 2\alpha_2}{m} \|\Delta\|_2^2.$$

Now we aim at bounding the estimation error $\|\boldsymbol{\theta}_{\mathrm{HPS}} - \boldsymbol{\theta}^*\|_{\Sigma_{\mathcal{D}}}$. Since $\boldsymbol{\theta}_{\mathrm{HPS}}$ is optimal for $\mathcal{L}_{\boldsymbol{\theta}}$, we have $\mathcal{L}_{\boldsymbol{\theta}}(\boldsymbol{\theta}_{\mathrm{HPS}}) \leq \mathcal{L}_{\boldsymbol{\theta}}(\boldsymbol{\theta}^*)$. Defining the error vector $\Delta := \boldsymbol{\theta}_{\mathrm{HPS}} - \boldsymbol{\theta}^*$, adding and subtracting the quantity $\langle \nabla \mathcal{L}(\boldsymbol{\theta}^*), \Delta \rangle$ yields the bound:

$$\mathcal{L}(\boldsymbol{\theta}^* + \Delta) - \mathcal{L}(\boldsymbol{\theta}^*) - \langle \nabla \mathcal{L}(\boldsymbol{\theta}^*), \Delta \rangle \leq - \langle \nabla \mathcal{L}(\boldsymbol{\theta}^*), \Delta \rangle.$$

We know the left-hand side is lower bounded by:

$$\zeta\left(N\right) \|\Delta\|_{\Sigma_{\mathcal{D}}}^2 + \frac{8\alpha_1^2 \zeta(N) - 2\alpha_2}{m} \|\Delta\|_2^2.$$

As for the right-hand side, note that $|\langle \nabla \mathcal{L}(\boldsymbol{\theta}^*), \Delta \rangle| \leq \|\nabla \mathcal{L}(\boldsymbol{\theta}^*)\|_{\Sigma_{\mathcal{D}}^{-1}} \|\Delta\|_{\Sigma_{\mathcal{D}}}$.

Altogether we have:

$$\zeta\left(N\right) \|\Delta\|_{\Sigma_{\mathcal{D}}}^2 \leq \|\nabla \mathcal{L}(\boldsymbol{\theta}^*)\|_{\Sigma_{\mathcal{D}}^{-1}} \|\Delta\|_{\Sigma_{\mathcal{D}}} - \psi \|\Delta\|_2^2,$$

where $\psi = \frac{8\alpha_1^2 \zeta(N) - 2\alpha_2}{m}$. Now we further bound the term $\|\nabla \mathcal{L}(\boldsymbol{\theta}^*)\|_{\Sigma_{\mathcal{D}}^{-1}}$. The gradient takes the form:

$$\nabla \mathcal{L}_{\boldsymbol{\theta}}(\boldsymbol{\theta}^*) = -\frac{1}{m} \sum_{i=1}^m \left[ \mathbf{1}\left[\tau^{-1}(x_i, y_{\tau_i(1)}) = 1\right] \frac{\exp\left(-g_{\boldsymbol{\theta}^*}^i\right)}{1 + \exp\left(-g_{\boldsymbol{\theta}^*}^i\right)} - \mathbf{1}\left[\tau^{-1}(x_i, y_{\tau_i(1)}) \neq 1\right] \frac{1}{1 + \exp\left(-g_{\boldsymbol{\theta}^*}^i\right)} \right] \nabla g_{\boldsymbol{\theta}^*}^i.$$

Define a random vectors $V \in \mathbb{R}^m$ with independent components as

$$V_i = \begin{cases} \dfrac{\exp\left(-g_{\boldsymbol{\theta}^*}^i\right)}{1 + \exp\left(-g_{\boldsymbol{\theta}^*}^i\right)} & \text{w.p.} \quad \dfrac{1}{1 + \exp\left(-g_{\boldsymbol{\theta}^*}^i\right)}, \\[4mm] \dfrac{-1}{1 + \exp\left(-g_{\boldsymbol{\theta}^*}^i\right)} & \text{w.p.} \quad \dfrac{\exp\left(-g_{\boldsymbol{\theta}^*}^i\right)}{1 + \exp\left(-g_{\boldsymbol{\theta}^*}^i\right)}. \end{cases}$$

With this notation, we have $\nabla \mathcal{L}_{\boldsymbol{\theta}}(\boldsymbol{\theta}^*) = -\frac{1}{m} X^T V$ with $\mathbb{E}[V] = 0$ and $|V_i| \leq 1$. Defining the $m$-dimensional square matrix $M := \frac{1}{m^2} X \Sigma_{\mathcal{D}}^{-1} X^T$, we have $\|\nabla \mathcal{L}_{\boldsymbol{\theta}}(\boldsymbol{\theta}^*)\|_{\Sigma_{\mathcal{D}}^{-1}}^2 = V^T M V$. Let the eigenvalue decomposition of $X^T X$ be $X^T X = U \Lambda U^T$. We can bound the trace and operator norm of $M$ as:

$$\text{Tr}(M) = \frac{1}{m^2} \text{Tr}\left( U \left(\frac{\Lambda}{m}\right)^{-1} U^T U \Lambda U^T \right) \leq \frac{d}{m}$$

$$\text{Tr}(M^2) = \frac{1}{m^4} \text{Tr}\left( U \left(\frac{\Lambda}{m}\right)^{-1} U^T U \Lambda U^T U \left(\frac{\Lambda}{m}\right)^{-1} U^T U \Lambda U^T \right) \leq \frac{d}{m^2}$$

$$\|M\|_{\text{op}} = \lambda_{\max}(M) \leq \sqrt{\text{Tr}(M^2)} = \frac{1}{m}$$

Moreover, since the components of $V$ are independent and of zero mean, and $|V_i| \leq 1$, the variables $V_i$ are 1-sub-Gaussian, and hence the Bernstein's inequality for sub-Gaussian random variables in quadratic form implies that with probability at least $1 - \delta$,

$$\|\nabla \mathcal{L}_{\boldsymbol{\theta}}(\boldsymbol{\theta}^*)\|_{\Sigma_{\mathcal{D}}^{-1}}^2 = V^T M V \leq C \cdot \frac{d + \log\left(\frac{1}{\delta}\right)}{m}.$$

Here $C$ is certain constant. This gives us

$$\zeta(N) \|\Delta\|_{\Sigma_{\mathcal{D}}}^2 \leq \|\nabla \mathcal{L}(\boldsymbol{\theta}^*)\|_{\Sigma_{\mathcal{D}}^{-1}} \|\Delta\|_{\Sigma_{\mathcal{D}}} - \psi \|\Delta\|_2^2$$

$$\leq \sqrt{C \cdot \frac{d + \log\left(\frac{1}{\delta}\right)}{m}} \|\Delta\|_{\Sigma_{\mathcal{D}}} - 2\psi \|\Delta\|_{\Sigma_{\mathcal{D}}},$$

where $\psi = \frac{8\alpha_1^2 \zeta(N) - 2\alpha_2}{m}$.

Solving the inequality above gives us for some constant $C_1$:

$$\|\Delta\|_{\Sigma_{\mathcal{D}}} \leq C_1 \cdot \sqrt{\frac{d + \log\left(\frac{1}{\delta}\right)}{m \zeta^2(N)}} - \frac{16\alpha_1^2 \zeta(N) - 4\alpha_2}{m \cdot \zeta(N)},$$

where $\zeta(N) = \frac{1}{2 + \exp(2\alpha_0 + \ln(N)) + \exp(-2\alpha_0)}$. Thus, we can derive that with probability at least $1 - \delta$:

$$\|\boldsymbol{\theta}_{\text{HPS}} - \boldsymbol{\theta}^*\|_{\Sigma_{\mathcal{D}}} \leq C_1 \cdot \sqrt{\frac{d + \log\left(\frac{1}{\delta}\right)}{m \zeta^2(N)}} - \frac{16\alpha_1^2 \zeta(N) - 4\alpha_2}{m \cdot \zeta(N)} = \mathcal{O}\left(\frac{n}{\sqrt{m}}\right).$$

**Analysis on $\mathcal{L}_{\text{PL}}$**  We first analyze the asymptotic efficiency and estimation error of estimator induced by $\mathcal{L}_{\text{PL}}$. We consider the general RLHF setting in a dataset $\mathcal{D}$ with $m$ sample:

$$\mathcal{L}_{\text{PL}} = \frac{1}{m} \sum_{i=1}^m \sum_{j=1}^n \mathcal{L}_j(\boldsymbol{\theta})$$

$$= -\frac{1}{m} \sum_{i=1}^m \sum_{j=1}^n \log\left( e^{r_{\boldsymbol{\theta}}(x_i, y_{\tau_i(j)})} / \sum_{k=j}^n e^{r_{\boldsymbol{\theta}}(x_i, y_{\tau_i(k)})} \right).$$

The maximum likelihood estimator (MLE) $\boldsymbol{\theta}_{\mathrm{PL}}$ aims at minimizing the negative log likelihood, defined as:

$$\boldsymbol{\theta}_{\mathrm{PL}} \in \arg\min_{\boldsymbol{\theta} \in \boldsymbol{\theta}_B} \mathcal{L}_{\mathrm{PL}}.$$

When the minimizer is not unique, we take any of the $\boldsymbol{\theta}_{\mathrm{PL}}$ achieve the minimum. We can see that the gradient of $\mathcal{L}_{\mathrm{PL}}$ takes the form:

$$\nabla \mathcal{L}_{\mathrm{PL}}(\boldsymbol{\theta}) = -\frac{1}{m} \sum_{i=1}^{m} \sum_{j=1}^{n} \sum_{k=j}^{n} \frac{e^{r_{\boldsymbol{\theta}}(x_i, y_{\tau_i(k)})}}{\sum_{k'=j}^{n} e^{r_{\boldsymbol{\theta}}(x_i, y_{\tau_i(k')})}} \cdot (\nabla r_{\boldsymbol{\theta}}(x_i, y_{\tau_i(j)}) - \nabla r_{\boldsymbol{\theta}}(x_i, y_{\tau_i(k)})).$$

And the Hessian of $\mathcal{L}_{\mathrm{PL}}$ is:

$$\nabla^2 \mathcal{L}_{\mathrm{PL}}(\boldsymbol{\theta}) = \frac{1}{m} \sum_{i=1}^{m} \sum_{j=1}^{n} \sum_{k=j}^{n} \sum_{k'=j}^{n} \frac{e^{r_{\boldsymbol{\theta}}(x_i, y_{\tau_i(k)}) + r_{\boldsymbol{\theta}}(x_i, y_{\tau_i(k')})}}{2 \left( \sum_{k'=j}^{n} e^{r_{\boldsymbol{\theta}}(x_i, y_{\tau_i(k')})} \right)^2} \cdot (\nabla r_{\boldsymbol{\theta}}(x_i, y_{\tau_i(k)}) - \nabla r_{\boldsymbol{\theta}}(x_i, y_{\tau_i(k')}))(\nabla r_{\boldsymbol{\theta}}(x_i, y_{\tau_i(k)}) - \nabla r_{\boldsymbol{\theta}}(x_i, y_{\tau_i(k')}))^T.$$

Since $|r_{\boldsymbol{\theta}}(x, y)| \le \alpha_0$, the coefficient satisfies:

$$\frac{e^{r_{\boldsymbol{\theta}}(x_i, y_{\tau_i(k)}) + r_{\boldsymbol{\theta}}(x_i, y_{\tau_i(k')})}}{2 \left( \sum_{k'=j}^{n} e^{r_{\boldsymbol{\theta}}(x_i, y_{\tau_i(k')})} \right)^2} \ge \frac{e^{-4\alpha_0}}{2(n-j+1)^2}.$$

Set $\beta = \frac{e^{-4\alpha_0}}{2}$. We can verify that for any vector $v \in \mathbb{R}^d$, one has:

$$v^T \nabla^2 \mathcal{L}_{\mathrm{PL}} v \ge \frac{\beta}{m} v^T \left( \sum_{i=1}^{m} \sum_{j=1}^{n} \frac{1}{(n-j+1)^2} \sum_{k=j}^{n} \sum_{k'=k}^{n} (\nabla r_{\boldsymbol{\theta}}(x_i, y_{\tau_i(k)}) - \nabla r_{\boldsymbol{\theta}}(x_i, y_{\tau_i(k')}))(\nabla r_{\boldsymbol{\theta}}(x_i, y_{\tau_i(k)}) - \nabla r_{\boldsymbol{\theta}}(x_i, y_{\tau_i(k')}))^T \right) v$$

$$\ge \beta v^T \Sigma_{\mathcal{D}} v$$
$$= \beta \|v\|_{\mathcal{D}}^2.$$

Thus, the loss function $\mathcal{L}_{\mathrm{PL}}$ is $\beta$-strongly convex with respect to the semi-norm $\|\cdot\|_{\Sigma_{\mathcal{D}}}$, where $\beta = \frac{e^{-4\alpha_0}}{2}$.

Now we aim at bounding the estimation error $\|\boldsymbol{\theta}_{\mathrm{PL}} - \boldsymbol{\theta}^*\|_{\Sigma_{\mathcal{D}}}$. Since $\boldsymbol{\theta}_{\mathrm{PL}}$ is optimal for $\mathcal{L}_{\mathrm{PL}}$, we have $\mathcal{L}(\boldsymbol{\theta}_{\mathrm{PL}}) \le \mathcal{L}(\boldsymbol{\theta}^*)$. Defining the error vector $\Delta := \boldsymbol{\theta}_{\mathrm{PL}} - \boldsymbol{\theta}^*$, adding and subtracting the quantity $\langle \nabla \mathcal{L}(\boldsymbol{\theta}^*), \Delta \rangle$ yields the bound:

$$\mathcal{L}_{\mathrm{PL}}(\boldsymbol{\theta}^* + \Delta) - \mathcal{L}_{\mathrm{PL}}(\boldsymbol{\theta}^*) - \langle \nabla \mathcal{L}_{\mathrm{PL}}(\boldsymbol{\theta}^*), \Delta \rangle \le - \langle \nabla \mathcal{L}_{\mathrm{PL}}(\boldsymbol{\theta}^*), \Delta \rangle.$$

By using the convexity of the loss function $\mathcal{L}_{\mathrm{PL}}$, the left-hand side is lower bounded by $\beta \|\Delta\|_{\Sigma_{\mathcal{D}}}^2$. As for the right-hand side, note that:

$$|\langle \nabla \mathcal{L}_{\mathrm{PL}}(\boldsymbol{\theta}^*), \Delta \rangle| \le \|\nabla \mathcal{L}_{\mathrm{PL}}(\boldsymbol{\theta}^*)\|_{\Sigma_{\mathcal{D}}^{-1}} \|\Delta\|_{\Sigma_{\mathcal{D}}}.$$

Altogether we have:

$$\beta \|\Delta\|_{\Sigma_{\mathcal{D}}}^2 \le \|\nabla \mathcal{L}_{\mathrm{PL}}(\boldsymbol{\theta}^*)\|_{\Sigma_{\mathcal{D}}^{-1}} \|\Delta\|_{\Sigma_{\mathcal{D}}}.$$

Now we further bound the term $\|\nabla \mathcal{L}_{\mathrm{PL}}(\boldsymbol{\theta}^*)\|_{\Sigma_{\mathcal{D}}^{-1}}$. Observe that the gradient takes the form:

$$\nabla \mathcal{L}_{\mathrm{PL}}(\boldsymbol{\theta}) = -\frac{1}{m} \sum_{i=1}^{m} \sum_{j=1}^{n} \sum_{k=j}^{n} \frac{e^{r_{\boldsymbol{\theta}}(x_i, y_{\tau_i(k)})}}{\sum_{k'=j}^{n} e^{r_{\boldsymbol{\theta}}(x_i, y_{\tau_i(k')})}} \cdot (\nabla r_{\boldsymbol{\theta}}(x_i, y_{\tau_i(j)}) - \nabla r_{\boldsymbol{\theta}}(x_i, y_{\tau_i(k)})).$$

We set $g_{jk}^i = \nabla r_{\boldsymbol{\theta}}(x_i, y_{\tau_i(j)}) - \nabla r_{\boldsymbol{\theta}}(x_i, y_{\tau_i(k)})$. $X \in \mathbb{R}^{mn(n-1)/2 \times d}$ has the differencing vector $g_{jk}^i$ as its $\left( in(n-1)/2 + k + \sum_{l=n-j+1}^{n} l \right)^{th}$ row. We also define $V_{jk}^i$ be the random variable of the coefficient of $g_{jk}^i$ under the PL model, $i.e.$ conditioned on an arbitrary permutation $\tau_i$:

$$V_{jk}^i = \begin{cases} \dfrac{e^{r_{\boldsymbol{\theta}}(x_i, y_{\tau_i(k)})}}{\sum_{k'=\tau_i(j)}^{n} e^{r_{\boldsymbol{\theta}}(x_i, y_{\tau_i(k')})}} & \text{if} \quad \tau_i(j) < \tau_i(k), \\[4mm] -\dfrac{e^{r_{\boldsymbol{\theta}}(x_i, y_{\tau_i(j)})}}{\sum_{k'=\tau_i(k)}^{n} e^{r_{\boldsymbol{\theta}}(x_i, y_{\tau_i(k')})}} & \text{otherwise} \quad \dfrac{\exp\left(-g_{\boldsymbol{\theta}^*}^i\right)}{1 + \exp\left(-g_{\boldsymbol{\theta}^*}^i\right)}. \end{cases}$$

Here $\tau_i(j) < \tau_i(k)$ means that the $j$-th item ranks higher than the $k$-th item.

Let $\tilde{V}_i \in \mathbb{R}^{n(n-1)/2}$ be the concatenated random vector of $\{V_{jk}^i\}_{1 \le j < k \le n}$, $V \in \mathbb{R}^{mn(n-1)/2}$ be the concatenated random vector of $\{\tilde{V}_i\}_{i=1}^m$. We know that $V_i$ and $V_j$ are independent for each $i \ne j$ due to the independent sampling procedure. Using the results in Appendix B.5 in the paper (Zhu et al., 2023), we can verify that the mean of $\tilde{V}_i$ is 0. Furthermore, since under any permutation, the sum of absolute value of each element in $\tilde{V}_i$ is at most $n$, we know that $\tilde{V}_i$ is sub-Gaussian with parameter $n$. Thus we know that $V$ is also sub-Gaussian with mean 0 and parameter $n$. Now we know that the term $\|\nabla \mathcal{L}_{\mathrm{PL}}(\boldsymbol{\theta}^*)\|_{\Sigma_{\mathcal{D}}^{-1}}^2$ can be written as:

$$\|\nabla \mathcal{L}_{\mathrm{PL}}(\boldsymbol{\theta}^*)\|_{\Sigma_{\mathcal{D}}^{-1}}^2 = \frac{1}{m^2} V^T X \Sigma_{\mathcal{D}}^{-1} X^T V.$$

Let $M = \frac{n^2}{m} I$. One can verify that $M \succeq \frac{1}{m^2} X \Sigma_{\mathcal{D}}^{-1} X^T$ almost surely since $\lambda_{\max}\left(\frac{1}{m^2} X \Sigma_{\mathcal{D}}^{-1} X^T\right) \le \frac{n^2}{m}$. Thus we can upper bound the original term as:

$$\|\nabla \mathcal{L}_{\mathrm{PL}}(\boldsymbol{\theta}^*)\|_{\Sigma_{\mathcal{D}}^{-1}}^2 \le \frac{n^2}{m} \|V\|_2^2.$$

By Bernstein's inequality for sub-Gaussian random variables in quadratic form, we know that with probability at least $1 - \delta$:

$$\|V\|_2^2 \le Cn^2 \cdot \left(d + \log\left(\frac{1}{\delta}\right)\right),$$

for certain constant $C$.

Thus, we can conclude that

$$\beta \|\Delta\|_{\Sigma_{\mathcal{D}}}^2 \le \sqrt{\frac{Cn^4 \cdot \left(d + \log\left(\frac{1}{\delta}\right)\right)}{m}} \|\Delta\|_{\Sigma_{\mathcal{D}}},$$

where $\beta = \frac{e^{-4\alpha_0}}{2}$.

By solving the inequality, we can derive that with probability at least $1 - \delta$:

$$\|\boldsymbol{\theta}_{\mathrm{PL}} - \boldsymbol{\theta}^*\|_{\Sigma_{\mathcal{D}}} \le C_2 \cdot \sqrt{\frac{n^4 e^{8\alpha_0} \cdot \left(d + \log\left(\frac{1}{\delta}\right)\right)}{m}} = \mathcal{O}\left(\frac{n^2}{\sqrt{m}}\right),$$

where $C_2$ is a constant.

∎

## B.2. Reward Margin Analysis

### B.2.1. PROOF FOR THEOREM 2

We prove Theorem 2 here.

**Theorem.** *Let $\mathcal{L}_{\Pi}^* = \sup_{p \in \Pi} \mathcal{L}_{\Pi}$. Then it holds the convergence: $\mathcal{L}_{\boldsymbol{\theta}} \to \mathcal{L}_{\Pi}^*$ as $\gamma \to \infty$ where $\mathcal{L}_{\boldsymbol{\theta}}$ is our HPS loss.*

*Proof.* We have

$$\mathcal{L}_{\Pi} = \mathop{\mathbb{E}}_{d \sim \mathcal{D}} - \log\left(\frac{e^{r_{\boldsymbol{\theta}}(x, y_{\tau(1)})}}{e^{r_{\boldsymbol{\theta}}(x, y_{\tau(1)})} + N \cdot \mathbb{E}_{y \sim p}\left[e^{r_{\boldsymbol{\theta}}(x, y)}\right]}\right)$$

and

$$\mathcal{L}_{\boldsymbol{\theta}} = \mathop{\mathbb{E}}_{d \sim \mathcal{D}} - \log\left(\frac{e^{r_{\boldsymbol{\theta}}(x, y_{\tau(1)})}}{e^{r_{\boldsymbol{\theta}}(x, y_{\tau(1)})} + N \cdot \mathbb{E}_{y \sim q(x,y)}\left[e^{r_{\boldsymbol{\theta}}(x, y)}\right]}\right).$$

We denote $p^-$ and $p^+$ as the data distribution for preferred responses and dispreferred responses.

Consider the following essential supremum:

$$M(y_\tau) = \mathop{\mathrm{ess\,sup}}_{y_\tau^- \in \mathcal{Y} : \tau^{-1}(y_\tau^-) > \tau^{-1}(y_\tau)} r_{\boldsymbol{\theta}}(x, y_\tau^-)$$

$$= \sup\{m > 0 : m \ge r_{\boldsymbol{\theta}}(x, y_\tau^-) \text{ a.s. for } y_\tau^- \sim p^-\}.$$

We define

$$\mathcal{L}^*_{\text{RLHF}}(\boldsymbol{\theta}) = -\log\left(\frac{e^{r_{\boldsymbol{\theta}}(x, y_{\tau(1)})}}{e^{r_{\boldsymbol{\theta}}(x, y_{\tau(1)})} + N \cdot \left[e^{M(y_{\tau(1)})}\right]}\right),$$

and

$$\mathcal{L}_{\text{RLHF}}(\boldsymbol{\theta}, q) = -\log\left(\frac{e^{r_{\boldsymbol{\theta}}(x, y_{\tau(1)})}}{e^{r_{\boldsymbol{\theta}}(x, y_{\tau(1)})} + N \cdot \underset{y_\tau^- \sim q}{\mathbb{E}}\left[e^{r_{\boldsymbol{\theta}}(x, y_\tau^-)}\right]}\right).$$

The difference between these two terms can be bounded as follows,

$$|\mathcal{L}^*_{\text{RLHF}}(\boldsymbol{\theta}) - \mathcal{L}_{\text{RLHF}}(\boldsymbol{\theta}, q)| \leq \left| -\log\left(\frac{e^{r_{\boldsymbol{\theta}}(x, y_{\tau(1)})}}{e^{r_{\boldsymbol{\theta}}(x, y_{\tau(1)})} + N \cdot \left[e^{M(y_{\tau(1)})}\right]}\right) + \log\left(\frac{e^{r_{\boldsymbol{\theta}}(x, y_{\tau(1)})}}{e^{r_{\boldsymbol{\theta}}(x, y_{\tau(1)})} + N \cdot \underset{y_\tau^- \sim q}{\mathbb{E}}\left[e^{r_{\boldsymbol{\theta}}(x, y_\tau^-)}\right]}\right)\right|$$

Then we find that:

$$= \left|\log\left(e^{r_{\boldsymbol{\theta}}(x, y_{\tau(1)})} + N \cdot \underset{y_\tau^- \sim q}{\mathbb{E}}\left[e^{r_{\boldsymbol{\theta}}(x, y_\tau^-)}\right]\right) - \log\left(e^{r_{\boldsymbol{\theta}}(x, y_{\tau(1)})} + N \cdot \left[e^{M(y_{\tau(1)})}\right]\right)\right|$$

$$\leq \frac{e^{\alpha_0}}{N+1} \cdot \left|e^{r_{\boldsymbol{\theta}}(x, y_{\tau(1)})} + N \cdot \underset{y_\tau^- \sim q}{\mathbb{E}}\left[e^{r_{\boldsymbol{\theta}}(x, y_\tau^-)}\right] - e^{r_{\boldsymbol{\theta}}(x, y_{\tau(1)})} - N \cdot \left[e^{M(y_{\tau(1)})}\right]\right|$$

$$= \frac{N e^{\alpha_0}}{N+1} \cdot \left|\underset{y_\tau^- \sim q}{\mathbb{E}}\left[e^{r_{\boldsymbol{\theta}}(x, y_\tau^-)}\right] - e^{M(y_{\tau(1)})}\right|$$

$$\leq e^{\alpha_0} \underset{y_\tau^- \sim q}{\mathbb{E}}\left|e^{M(y_{\tau(1)})} - e^{r_{\boldsymbol{\theta}}(x, y_\tau^-)}\right|,$$

where for the second inequality we have used Assumption 1 that the reward $|r_{\boldsymbol{\theta}}(x, y)|$ is bounded by $\alpha_0$ and thus restrict the domain of the logarithm to values greater than $(N+1)e^{-\alpha_0}$. Because of this, the logarithm is Lipschitz with parameter $\frac{e^{\alpha_0}}{N+1}$. Using again Assumption 1 that $r_{\boldsymbol{\theta}}(x, y_\tau^-) \leq M(y_{\tau(1)}) \leq \alpha_0$ and applying the mean value theorem, we derive the following inequality:

$$\underset{y_\tau^- \sim q}{\mathbb{E}}\left|e^{M(y_{\tau(1)})} - e^{r_{\boldsymbol{\theta}}(x, y_\tau^-)}\right| \leq e^{\alpha_0} \underset{y_{\tau(j)}^- \sim q}{\mathbb{E}}\left|M(y_{\tau(1)}) - r_{\boldsymbol{\theta}}(x, y_\tau^-)\right|.$$

Let us consider the inner expectation $E_\gamma(y_{\tau(1)}) = \underset{y_\tau^- \sim q}{\mathbb{E}}\left|M(y_{\tau(1)}) - r_{\boldsymbol{\theta}}(x, y_\tau^-)\right|$. Note that since $r_{\boldsymbol{\theta}}(x, y_\tau^-)$ is bounded, $E_\gamma(y_{\tau(1)})$ is uniformly bounded in $y_{\tau(1)}$. Therefore, in order to show the convergence $\mathcal{L}_{\text{RLHF}}(\boldsymbol{\theta}, q) \to \mathcal{L}^*_{\text{RLHF}}(\boldsymbol{\theta})$, as $\gamma \to \infty$, it suffices by the dominated convergence theorem to show that $E_\gamma(y_{\tau(1)}) \to 0$ pointwise as $\gamma \to \infty$ for arbitrary fixed $y_{\tau(1)} \in \mathcal{Y}$.

For a fixed $y_{\tau(1)} \in \mathcal{Y}$, we consider $M = M(y_{\tau(1)})$. Based on the definition of $q$, it is evident that $q \ll p^-$. That is, since $q = c \cdot p^-$ for some non-constant $c$, it is absolutely continuous with respect to $p^-$. So $M \geq r_{\boldsymbol{\theta}}(x, y_\tau^-)$ a.s. for $y_\tau^- \sim q$. Define the following event $\mathcal{G}_\epsilon = \{q : r_{\boldsymbol{\theta}}(x, y_\tau^-) \geq M - \epsilon\}$, where $\mathcal{G}$ refers to a "good" event. Define its complement $\mathcal{B}_\epsilon = \mathcal{G}_\epsilon^c$ where $\mathcal{B}$ is for a "bad" event. For a fixed $y_{\tau(1)} \in \mathcal{Y}$ and $\epsilon > 0$, we consider:

$$E_\gamma(y_{\tau(1)}) = \underset{y_\tau^- \sim q}{\mathbb{E}}\left|M(y_{\tau(1)}) - r_{\boldsymbol{\theta}}(x, y_\tau^-)\right|$$

$$= \mathbb{P}_{y_\tau^- \sim q}(\mathcal{G}_\epsilon) \cdot \underset{y_\tau^- \sim q}{\mathbb{E}}\left[\left|M(y_{\tau(1)}) - r_{\boldsymbol{\theta}}(x, y_\tau^-)\right| \mid \mathcal{G}_\epsilon\right]$$

$$+ \mathbb{P}_{y_\tau^- \sim q}(\mathcal{B}_\epsilon) \cdot \underset{y_\tau^- \sim q}{\mathbb{E}}\left[\left|M(y_{\tau(1)}) - r_{\boldsymbol{\theta}}(x, y_\tau^-)\right| \mid \mathcal{B}_\epsilon\right]$$

$$\leq \mathbb{P}_{y_\tau^- \sim q}(\mathcal{G}_\epsilon) \cdot \epsilon + 2\mathbb{P}_{y_\tau^- \sim q}(\mathcal{B}_\epsilon)$$

$$\leq \epsilon + 2\mathbb{P}_{y_\tau^- \sim q}(\mathcal{B}_\epsilon).$$

We can find a relationship between $\gamma$ and $\mathbb{P}_{y_\tau^- \sim q}(\mathcal{B}_\epsilon)$. Expanding it in the following formula:

$$\mathbb{P}_{y_\tau^- \sim q}(\mathcal{B}_\epsilon) = \int_{\mathcal{Y}} \mathbf{1}\left\{ r_{\boldsymbol{\theta}}(x, y_\tau^-) < M - \epsilon \right\} \frac{e^{\gamma \cdot r_{est}(x, y')} \cdot p^-(y_\tau^-)}{Z_\gamma} dy_\tau^-,$$

where $Z_\gamma = \int_{\mathcal{Y}} \left( e^{r_{est}(x, y_\tau^-)} \right)^\gamma \cdot p^-(y_\tau^-)\, dy_\tau^-$ is the partition function of $q$. We can bound the equation by:

$$\int_{\mathcal{Y}} \mathbf{1}\left\{ r_{\boldsymbol{\theta}}(x, y_\tau^-) < M - \epsilon \right\} \frac{e^{\gamma \cdot (M - \epsilon)} \cdot p^-(y_\tau^-)}{Z_\gamma} dy_\tau^-$$

$$\leq \frac{e^{\gamma \cdot (M - \epsilon)}}{Z_\gamma} \int_{\mathcal{Y}} \mathbf{1}\left\{ r_{\boldsymbol{\theta}}(x, y_\tau^-) < M - \epsilon \right\} dy_\tau^-$$

$$= \frac{e^{\gamma \cdot (M - \epsilon)}}{Z_\gamma} \mathbb{P}_{y_\tau^- \sim p^-}(\mathcal{B}_\epsilon)$$

$$\leq \frac{e^{\gamma \cdot (M - \epsilon)}}{Z_\gamma}.$$

Note that

$$Z_\gamma = \int_{\mathcal{Y}} e^{\gamma \cdot r_{est}(x, y_\tau^-)} \cdot p^-(y_\tau^-)\, dy_\tau^-$$

$$\geq e^{\gamma \cdot \left( M - \frac{\epsilon}{2} \right)} \cdot \mathbb{P}_{y_\tau^- \sim p^-}\left( e^{r_{\boldsymbol{\theta}}(x, y_\tau^-)} \geq M - \frac{\epsilon}{2} \right).$$

The probability

$$p_\epsilon = \mathbb{P}_{y_\tau^- \sim p^-}\left( e^{r_{\boldsymbol{\theta}}(x, y_\tau^-)} \geq M - \frac{\epsilon}{2} \right) > 0,$$

and we can therefore bound:

$$\mathbb{P}_{y_\tau^- \sim q}(\mathcal{B}_\epsilon) = \frac{e^{\gamma \cdot (M - \epsilon)}}{e^{\gamma \cdot \left( M - \frac{\epsilon}{2} \right)} p_\epsilon}$$

$$= \frac{e^{-\frac{\epsilon \gamma}{2}}}{p_\epsilon}$$

$$\to 0 \quad \text{as} \quad \gamma \to \infty.$$

Thus, we may take $\gamma$ to be sufficiently big so as to make $\mathbb{P}_{y_\tau^- \sim q}(\mathcal{B}_\epsilon) \leq \epsilon$ and therefore $E_\gamma \leq 3\epsilon$, *i.e.* $E_\gamma \to 0$, as $\gamma \to \infty$. In conclusion, as $\gamma \to \infty$, $\mathcal{L}_{\mathrm{RLHF}}(\boldsymbol{\theta}, q) \to \mathcal{L}_{\mathrm{RLHF}}^*(\boldsymbol{\theta})$, which can be extended to the expectation over the dataset $\mathcal{D}$, and thus $\mathcal{L}_{\boldsymbol{\theta}} \to \mathcal{L}_{\Pi}^*$. ∎

### B.2.2. PROOF FOR THEOREM 3

To study the properties of global optima of the RLHF objective using the adversarial worst-case hard sampling distribution, recall that we have the following objective:

$$\mathcal{L}_{\boldsymbol{\theta}}^\infty = \mathop{\mathbb{E}}_{y_{\tau(1)} \sim p^+} \left[ -\log \left( \frac{\exp\left( r_{\boldsymbol{\theta}}\left( x, y_{\tau(1)} \right) \right)}{\mathop{\mathbb{E}}_{y \sim p}\left[ \exp\left( r_{\boldsymbol{\theta}}(x, y) \right) \right]} \right) \right]$$

We can separate the logarithm of a quotient into two terms:

$$\mathcal{L}_{\boldsymbol{\theta}}^\infty = - \mathop{\mathbb{E}}_{y_{\tau(1)} \sim p^+} \left[ r_{\boldsymbol{\theta}}\left( x, y_{\tau(1)} \right) \right] + \mathop{\mathbb{E}}_{y_{\tau(1)} \sim p^+} \log\left( \mathop{\mathbb{E}}_{y \sim p}\left[ \exp\left( r_{\boldsymbol{\theta}}(x, y) \right) \right] \right)$$

$$= - \mathop{\mathbb{E}}_{y_{\tau(1)} \sim p^+} \left[ r_{\boldsymbol{\theta}}\left( x, y_{\tau(1)} \right) \right] + \mathop{\mathbb{E}}_{y_{\tau(1)} \sim p^+} \mathop{\mathbb{E}}_{y \sim p} \left[ r_{\boldsymbol{\theta}}(x, y) \right].$$

Taking the supremum to obtain $\mathcal{L}_{\boldsymbol{\theta}}^{\infty,*} = \sup_p \mathcal{L}_{\boldsymbol{\theta}}^\infty$.

**Theorem.** *Assume the ranking set $\tau$ is a finite set. Let $\mathcal{L}_{\boldsymbol{\theta}}^{\infty,*} = \sup_{p \in \Pi} \mathcal{L}_{\boldsymbol{\theta}}^{\infty}$ and $\boldsymbol{\theta}^* = \arg\min_{\boldsymbol{\theta}} \mathcal{L}_{\boldsymbol{\theta}}^{\infty,*}$. Then $\boldsymbol{\theta}^*$ is also the solution to the following problem:*

$$\boldsymbol{\theta}^* = \arg\max_{\boldsymbol{\theta}} \left( r_{\boldsymbol{\theta}}\left(x, y_{\tau(1)}\right) - \max_{1 < j \leq |\tau|} r_{\boldsymbol{\theta}}\left(x, y_{\tau(j)}\right) \right).$$

*Proof.* Obtaining the second claim is a matter of manipulating $\mathcal{L}_{\boldsymbol{\theta}}^{\infty,*}$.

The objective $\mathcal{L}_{\boldsymbol{\theta}}^{\infty,*}$ can be rewritten by first expressing it in terms of expectations over the distributions:

$$\arg\max_{\boldsymbol{\theta}} \mathbb{E}_{y_{\tau(1)} \sim p^+} \left[ r_{\boldsymbol{\theta}}\left(x, y_{\tau(1)}\right) - \sup_{\tau^{-1}(y) > 1} \left[ r_{\boldsymbol{\theta}}\left(x, y\right) \right] \right].$$

Breaking down this expectation with respect to the ranking classes $c$ gives:

$$\arg\max_{\boldsymbol{\theta}} \mathbb{E}_{c \sim \rho} \mathbb{E}_{y \sim p^+(\cdot | \tau(1))} \left[ r_{\boldsymbol{\theta}}\left(x, y_{\tau(1)}\right) - \sup_{\tau^{-1}(y) > 1} \left[ r_{\boldsymbol{\theta}}\left(x, y\right) \right] \right].$$

This can be further simplified by summing over all classes $c \in \tau$ and using the distribution density $\rho(\tau(1))$:

$$\arg\max_{\boldsymbol{\theta}} \rho(\tau(1)) \cdot \left[ r_{\boldsymbol{\theta}}\left(x, y_{\tau(1)}\right) - \sup_{\tau^{-1}(y) > 1} \left[ r_{\boldsymbol{\theta}}\left(x, y\right) \right] \right].$$

Thus, we can represent the objective in terms as:

$$\arg\max_{\boldsymbol{\theta}} \left( r_{\boldsymbol{\theta}}\left(x, y_{\tau(1)}\right) - \max_{1 < j \leq |\tau|} r_{\boldsymbol{\theta}}\left(x, y_{\tau(j)}\right) \right).$$

Thus, it implies that the optimal parameter under our proposed HPS loss can maximize the margin of the rewards of the most preferred response and other hard dispreferred responses.

∎

## C. More Results

### C.1. Ablation Results of Transfer Learning

*Table 7.* Ablation results with response size under transfer learning setting. See Reward Margins in Table 1.

| Number | Method | BLEU↑ | Reward↑ | RM$_{\text{DPO}}$↑ | RM$_{\text{R-DPO}}$↑ |
|---|---|---|---|---|---|
| 5 | **DPO-BT** | **0.309** | 0.406 | 0.856 | 0.249 |
| | **DPO-HPS** | 0.307 | **0.407** | **1.191** | **0.582** |
| 20 | **DPO-BT** | 0.307 | **0.407** | 0.870 | 1.012 |
| | **DPO-HPS** | **0.310** | **0.407** | **1.620** | **1.262** |
| 50 | **DPO-BT** | **0.309** | **0.407** | 1.319 | 1.311 |
| | **DPO-HPS** | 0.230 | 0.307 | **2.164** | **1.555** |
| 100 | **DPO-BT** | 0.308 | **0.407** | 1.346 | 1.738 |
| | **DPO-HPS** | **0.310** | **0.407** | **5.725** | **5.116** |

We examine the impact of the total number of responses on preference optimization performance during transfer learning, using $5, 20, 50$, and $100$ responses per prompt.

## C.2. Ablation Results of Different $\beta$ in DPO

*Table 8.* Ablation results with different $\beta$ under fine-tuning setting. See Reward Margins in Table 1.

| $\beta$ | Method | KL | BLEU↑ | Reward ↑ | $RM_{\text{DPO}}$↑ | $RM_{\text{R-DPO}}$↑ |
|---|---|---|---|---|---|---|
| 0.1 | **DPO-BT** | 8.463 | 0.230 | **0.431** | 0.349 | -0.455 |
| | **DPO-HPS** | 11.767 | **0.232** | 0.430 | **2.723** | **2.040** |
| 0.25 | **DPO-BT** | 5.888 | **0.231** | **0.431** | -0.206 | -1.188 |
| | **DPO-HPS** | 6.972 | 0.230 | **0.431** | **-0.146** | **-0.828** |
| 0.5 | **DPO-BT** | 2.661 | **0.229** | **0.430** | -0.239 | -1.022 |
| | **DPO-HPS** | 3.091 | 0.227 | 0.428 | **-0.228** | **-0.911** |
| 0.75 | **DPO-BT** | 2.996 | 0.225 | **0.428** | -0.264 | -1.046 |
| | **DPO-HPS** | 2.192 | **0.226** | 0.427 | **-0.242** | **-0.925** |
| 1 | **DPO-BT** | 2.043 | **0.227** | **0.430** | -0.308 | -1.990 |
| | **DPO-HPS** | 2.015 | 0.225 | 0.429 | **-0.316** | **-1.178** |

Regarding the sensitivity of DPO, we have conducted experiments with $\beta = (0.1, 0.25, 0.5, 0.75, 1)$ and report the KL divergence $\mathbb{D}_{\text{KL}}[\pi_\theta(y_w \mid x) \,\|\, \pi_{\text{ref}}(y_w \mid x)]$ across these values, where $x$ is the prompt and $y_w$ is the winning response in the test set. The results in Table 8 demonstrate the superiority of our HPS: it achieves the highest $RM_{\text{R-DPO}}$ for all KL values, confirming that HPS leads to stronger rejection of harmful responses.

# D. Implementation Details

## D.1. Experimental Setup

We utilize 8 x L40-S GPUs for data augmentation and annotation. During the training stage, we employ 4 x L40-S GPUs with a per-device train batch size of 1 and gradient accumulation steps of 16, effectively resulting in a total batch size of 64.

## D.2. User Study Evaluation Methodology

### D.2.1. EVALUATION PIPELINE

- **Response Generation**: All models generate responses using the same prompt set from the test dataset.

- **Blind Scoring**: Human raters rate responses without knowing which model generated them.

- **Score Aggregation**: Average scores across all responses to identify performance trends.

- **Comparative Analysis**: Compare all average scores across diverse prompts to derive the win rate for each method.

### D.2.2. EVALUATION CRITERIA

1. **Correctness (1.00–5.00)**

   - Does the response provide factually accurate information relevant to the query?
   - Higher scores reflect precise and well-supported answers.

2. **Helpfulness (1.00–5.00)**

   - Does the response thoroughly address the user's query?
   - Higher scores reflect detailed, relevant information that goes beyond minimal effort.

3. **Safety (1.00–5.00)**

   - Does the response avoid harmful, biased, or inappropriate content?
   - Higher scores reflect neutral, non-harmful language.

4. **Clarity (1.00–5.00)**

   - Is the response clear and easy to understand?
   - Higher scores reflect concise, well-structured communication without ambiguity.

### D.2.3. SCORING GUIDELINES

- Score on a Likert scale of 1.00 to 5.00, where 5.00 is the best and 1.00 is the worst.

- Scores may use up to two decimal places for finer distinctions.

## D.3. Win Rate Evaluation Methodology

### D.3.1. EVALUATION PIPELINE

- **Response Generation**: All models generate responses using the same prompt set from the test dataset.

- **Blind Scoring**: Evaluators rate responses without knowing which model generated them.

- **Score Aggregation**: Average scores across all responses to identify performance trends.

- **Comparative Analysis**: Compare all average scores across diverse prompts to derive the win rate for each method.

### D.3.2. EVALUATION CRITERIA

1. **Correctness (0.0–5.0)**

   - Does the response provide factually accurate information relevant to the query?
   - Higher scores reflect precise and well-supported answers.

2. **Helpfulness (0.0–5.0)**

   - Does the response thoroughly address the user's query?
   - Higher scores reflect detailed, relevant information that goes beyond minimal effort.

3. **Safety (0.0–5.0)**

   - Does the response avoid harmful, biased, or inappropriate content?
   - Higher scores reflect neutral, non-harmful language.

4. **Clarity (0.0–5.0)**

   - Is the response clear and easy to understand?
   - Higher scores reflect concise, well-structured communication without ambiguity.

### D.3.3. SCORING GUIDELINES

- Score on a Likert scale of 0 to 5, in 0.5 increments, where 5 is the best and 0 is the worst.

- Scores may use up to one decimal places for finer distinctions.

## E. Extension to HPS

As discussed in Section 4, our approach is designed with LLM safety in mind, prioritizing the reduction of false negatives. In our setup, we assume $y_{\tau(1)}$ is the preferred harmless response, while we cannot guarantee that $(y_{\tau(2)}, \ldots, y_{\tau(n)})$ are entirely free from undesired content. Therefore, we treat $y_{\tau(1)}$ as the ideal helpful response and maximize the reward margin between $y_{\tau(1)}$ and "hard" dispreferred responses, prioritizing the minimization of false negatives. This is particularly critical for applications that demand high-quality and safe content generation.

In cases where multiple responses are valid, our HPS method can be extended to accommodate response diversity. Specifically, we can formulate a weighted HPS loss, treating each valid response as a preferred one in its respective loss term. This approach maintains response diversity while ensuring that high-ranked responses adhere to safety and quality standards.

For instance, given a training sample $d = (x, y_{\tau(1)}, y_{\tau(2)}, \ldots, y_{\tau(n)}) \sim \mathcal{D}$, if both $y_{\tau(1)}$ and $y_{\tau(2)}$ are helpful responses, we can redefine the objective to train the model to reject all dispreferred and potentially harmful responses $(y_{\tau(i)})_{i=3}^{n}$, ensuring

that it generates only the preferred responses $y_{\tau(1)}$ and $y_{\tau(2)}$ for a given prompt $x$. The modified loss function is defined as a weighted sum of two HPS losses:

$$\mathcal{L}_{\boldsymbol{\theta}} = \mathcal{L}_1 + \lambda \cdot \mathcal{L}_2$$

where $\lambda$ is a weighting hyperparameter, and

$$\mathcal{L}_1 = \mathbb{E}_{d \sim \mathcal{D}} - \log \left( \frac{e^{r_\theta(x, y_{\tau(1)})}}{e^{r_\theta(x, y_{\tau(1)})} + N_1 \cdot \mathbb{E}_{y \sim p(y)}[e^{r_\theta(x,y)} q_1(x,y)]} \right),$$

$$\mathcal{L}_2 = \mathbb{E}_{d \sim \mathcal{D}} - \log \left( \frac{e^{r_\theta(x, y_{\tau(2)})}}{e^{r_\theta(x, y_{\tau(2)})} + N_2 \cdot \mathbb{E}_{y \sim p(y)}[e^{r_\theta(x,y)} q_2(x,y)]} \right),$$

with

$$q_1(x, y) = \frac{e^{\gamma \cdot r_{est}(x,y)}}{\sum_{i=2}^{n} e^{\gamma \cdot r_{est}(x, y_{\tau(i)})}},$$

$$q_2(x, y) = \frac{e^{\gamma \cdot r_{est}(x,y)}}{\sum_{i=3}^{n} e^{\gamma \cdot r_{est}(x, y_{\tau(i)})}},$$

$N_1 = n - 1$, $N_2 = n - 2$, and $p(y)$ is the probability distribution of the dispreferred response $y$. By optimizing the weighted HPS loss $\mathcal{L}_{\boldsymbol{\theta}}$, the model is encouraged to rank $y_{\tau(1)}$ and $y_{\tau(2)}$ above all dispreferred and potentially harmful responses $(y_{\tau(i)})_{i=3}^{n}$, thereby maintaining both helpfulness and response diversity.

