# OpenReview forum: "HPS: Hard Preference Sampling for Human Preference Alignment"
_ICML.cc/2025/Conference — ICML 2025 poster_

### Official Review · Reviewer_85yJ · 2025-03-10

**Overall Recommendation:** 2

**Summary:**

The paper introduces Hard Preference Sampling (HPS), a framework for aligning large language models with human preferences. Traditional methods face challenges with harmful content, inefficient use of dispreferred responses, and high computational costs. HPS addresses these issues through a training loss that prioritizes preferred responses while rejecting dispreferred ones, with special emphasis on "hard" dispreferred samples that closely resemble preferred ones to enhance the model’s rejection capabilities.

**Claims And Evidence:**

The paper assumes the first response is preferred while subsequent responses are potentially harmful or purely dispreferred. However, this overlooks preference diversity among humans—several top-ranked responses might be acceptable but with varying preference levels based on individual tastes regarding grammar, expression style, and other subjective factors, rather than being inherently harmful.

**Essential References Not Discussed:**

The paper should discuss alignment methods based on list responses like [1,2].

[1] LiPO: Listwise Preference Optimization through Learning-to-Rank.
[2] SLiC-HF: Sequence Likelihood Calibration with Human Feedback.

**Experimental Designs Or Analyses:**

1. The paper mentions "sampling only a single importance-weighted dispreferred response" but fails to clarify the specific selection method—whether random or based on certain criteria. Additionally, the experiments could have benefited from comparing different sampling strategies to strengthen their claims.
2. The authors use Qwen-2.5 for evaluation but fail to specify which variant (3B, 7B, or 72B). They also don't justify why they chose Qwen over more commonly used models like GPT or Claude, which are standard in most preference alignment papers.
3. The authors only compare their approach with PL model-based methods, overlooking alternative alignment techniques such as Lambda loss [1], which could have provided a more comprehensive evaluation of their method's effectiveness.

[1] LiPO: Listwise Preference Optimization through Learning-to-Rank

**Methods And Evaluation Criteria:**

Yes

**Other Comments Or Suggestions:**

1. The paper's writing quality could be improved, as it contains several repetitive statements and fails to properly define all variables in the theorems, such as $d$ in Theorem 1.

**Other Strengths And Weaknesses:**

Strengths:
1. HPS introduces a training loss that explicitly prioritizes preferred responses while rejecting all dispreferred ones, focusing particularly on "hard" negative examples.
2. Experiments show substantial improvements in reward margins compared to traditional methods.
3. HPS innovatively applies Monte Carlo importance sampling to replace the dispreferred term in PL loss, offering a more efficient
alignment approach.

Weaknesses:
1. The paper assumes the first response is preferred while subsequent responses are potentially harmful or purely dispreferred. However, this overlooks preference diversity among humans—several top-ranked responses might be acceptable but with varying preference levels based on individual tastes regarding grammar, expression style, and other subjective factors, rather than being inherently harmful.
2. In the implementation, the authors sample only a single importance-weighted dispreferred response, causing the method to degenerate into standard Bradley-Terry model. This design choice naturally results in faster computation compared to PL model-based methods.
3. Since experiments only involve $n \leq 100$, the difference between the two error bounds amounts to merely a constant factor, suggesting the authors may have overstated the theoretical contribution.

**Questions For Authors:**

1. The paper mentions "sampling only a single importance-weighted dispreferred response" but fails to clarify the specific selection method—whether random or based on certain criteria. Additionally, the experiments could have benefited from comparing different sampling strategies to strengthen their claims.
2. The authors use Qwen-2.5 for evaluation but fail to specify which variant (3B, 7B, or 72B). They also don't justify why they chose Qwen over more commonly used models like GPT or Claude, which are standard in most preference alignment papers.
3. The authors only compare their approach with PL model-based methods, overlooking alternative alignment techniques such as Lambda loss [1], which could have provided a more comprehensive evaluation of their method's effectiveness.


[1] LiPO: Listwise Preference Optimization through Learning-to-Rank

**Relation To Broader Scientific Literature:**

The paper should discuss alignment methods based on list responses like [1,2].

[1] LiPO: Listwise Preference Optimization through Learning-to-Rank.
[2] SLiC-HF: Sequence Likelihood Calibration with Human Feedback.

**Theoretical Claims:**

I briefly examine the theoretical proofs.

---

> ### Author Rebuttal · Authors · 2025-04-01
>
> Thank you for the insightful comments! We provide our response and hope our response addresses your concerns. We also look forward to the subsequent discussion which may further help solve the current issues.
>
> **1) Our revision will discuss alignment methods like SLiC-HF (arXiv:2305.10425) and LiPO (arXiv:2402.01878) based on list responses. SLiC-HF** is an alternative to RLHF-PPO by integrating the sequence-level contrastive method SLiC (arXiv:2210.00045) with human preference rankings:
> $$
> \mathcal{L}(\theta)=\max(0,\delta-\log(\pi\_{\theta}(y^{+}|x))+\log(\pi\_{\theta}(y^{-}|x))-\lambda\log(\pi\_{\theta}(y\_{ref}|x))).
> $$
> $y^{+}$, $y^{-}$, and $y_{ref}$ denote the positive, negative, and reference sequences, respectively. $\delta$ is a margin hyperparameter and $\lambda$ is a regularization weight. In contrast, our HPS framework focuses on rejecting all potentially harmful responses while leveraging the varying informativeness of dispreferred responses.
>
> In **LiPO-$\lambda$**, it employs a listwise ranking objective with a Lambda weight $\Delta_{i,j}$. Given a list of responses $\boldsymbol{y}=(y_1,\dots,y_K)$,$$\mathcal{L}\_{LiPO}=\mathbb{E}\_{(x,\boldsymbol{y},\psi)\sim\mathcal{D}}\left[\sum\_{\psi\_{i}>\psi\_{j}}\Delta\_{i,j}\log(1+e^{-(s\_i-s\_j)})\right],$$where $\Delta_{i,j}=|2^{\psi_i}-2^{\psi_j}|\cdot|\frac{1}{\log(1+\tau(i))}-\frac{1}{\log(1+\tau(j))}|.$
> Here, $\psi_{i}$ is the true reward score of response $y_i$, and $s_i=\beta\log\frac{\pi_{\theta}(y_i|x)}{\pi_{{ref}}(y_i|x)}$ is the implicit DPO reward. The rank position of $y_i$ in the ordering induced by $\mathbf{s}=(s_1,\dots,s_K)$ is denoted as $\tau(i)$. The Lambda weight assigns greater importance to response pairs with larger preference gaps, i.e., $\psi_i-\psi_j$. However, our HPS prioritizes *hard* dispreferred responses—those that closely resemble the correct output but remain incorrect.
>
> **2) Our HPS can be extended to the setting where multiple top responses are valid.** Please see our response to Reviewer 674W.
>
> **3) Sampling a single importance-weighted dispreferred response DOES NOT reduce HPS to standard BT**, since HPS designs an importance-weighted sampling strategy, unlike BT’s deterministic selection.
>
> In HPS, a dispreferred response is sampled based on the importance-weighted distribution (L244):$$q(x,y)=\frac{e^{\gamma\cdot r_{est}(x,y)}}{\sum_{i=2}^{n}e^{\gamma\cdot r_{est}(x,y_{\tau(i)})}},\tag{1}$$where $(y_{\tau(i)})_{i=2}^{n}$ denote the dispreferred responses of a prompt $x$. This ensures that harder dispreferred responses—those more challenging to distinguish from preferred ones—are sampled more frequently and penalized more during training.
>
> In contrast, BT always selects a fixed pair: the most preferred and the most dispreferred response, ignoring all other dispreferred responses. However, the most dispreferred response is often the easiest to reject. HPS addresses this by prioritizing harder responses, enabling the model to refine its distinctions between preferred and dispreferred outputs.
>
> Empirically, BT struggles to capture the preference gap effectively. Table 2 shows that compared to BT, HPS significantly improves reward margins, reducing detrimental responses.
>
> **4)** While our experiments focus on $n\leq 100$, Thm. 1 analytically characterizes how error bounds $\Psi_{1}$ and $\Psi_{2}$ scale with $n$, providing insights for larger settings. We limit $n$ to 100 due to limited GPU resources.
>
> Moreover, Table 5 shows that as $n$ increases, the reward margin metrics consistently improve, aligning with the theoretical scaling behavior.
>
> **5)** In Thm. 1, $d$ is defined in Assumption 1 (L239), and denotes the parameter dimension of the reward model. For implicit reward modeling, it corresponds to the trainable LLM’s parameter dimension. We will carefully refine our manuscript.
>
> **6)** HPS samples a single dispreferred response via the importance-weighted distribution $q(x,y)$ (see Eq. 1 in response **3)**)
>
> $q(x,y)$ defines a probability distribution over these responses, so sampling strictly follows this distribution. Deviating from it would introduce a different approach, compromising the theoretical guarantees in Sec. 5, particularly *HPS’s improved sample efficiency and reward margin maximization over PL methods.*
>
> **7)** We use Qwen2.5-72B-Instruct for evaluation. In Sec. 7, we acknowledge that budget constraints limit us to open-source LLMs for estimating win rates. To strengthen our evaluation, we conducted a user study with human participants. See Tab.2 in our response **2)** to Reviewer v4s5 for details.
>
> **8)** Thanks. Since we mainly analyze PL and BT, we use these methods to investigate theoretical implications and empirical performance. To address your concern, we compare HPS with LiPO-$\lambda$ on HH-RLHF and find that HPS significantly improves the Reward Margin, limiting harmful responses.
>
> |Method|BLEU|Reward|$RM_{DPO}$|$RM_{R-DPO}$|
> |-|-|-|-|-|
> |**LiPO-$\lambda$**|0.229|0.430|1.437|1.121|

---

### Official Review · Reviewer_674W · 2025-03-14

**Overall Recommendation:** 3

**Summary:**

This paper propose a novel HPS method to prioritize the most preferred response while rejecting all other responses.

**Claims And Evidence:**

Yes.

**Essential References Not Discussed:**

No.

**Experimental Designs Or Analyses:**

The author mainly set three experiment sets:

1. Main experiments comparing HPS with naive PL and BT modeling method.
2. Human evaluation comparing HPS with the SFT/DPO-BT/DPO-PL baselines.
3. Ablation on response number under the fine-tuning setting.

**Methods And Evaluation Criteria:**

Yes.

**Other Comments Or Suggestions:**

1. The author propose to reject all but the top-1 ranked response, which might not be appropriate for a diverse range of scenarios where multiple possible responses are all good.

**Other Strengths And Weaknesses:**

Strength
1. The proposed HPS achieves better performance compared with PL and BT baseline.

**Questions For Authors:**

No.

**Relation To Broader Scientific Literature:**

The proposed method could be helpful for the field of LLM alignment targeting more helpful and harmless AI.

**Theoretical Claims:**

Yes, I checked the sampling complexity and reward margin analysis.

---

> ### Author Rebuttal · Authors · 2025-04-01
>
> Thank you for the insightful and positive comments! In the following, we provide our point-by-point response and hope our response helps address your concerns. We also look forward to the subsequent discussion which may further help solve the current issues.
>
> **Our HPS method can also be extended to the setting where multiple top responses are all valid.** As stated in Sec 4 (L138), our primary objective is to ensure that models generate helpful and harmless responses while avoiding harmful or dispreferred outputs. In our setup, we assume $y\_{\tau(1)}$ is the preferred harmless response, while we cannot guarantee that $(y\_{\tau(2)},\dots,y\_{\tau(n)})$ are entirely free from undesired content. Therefore, we treat $y\_{\tau(1)}$ as the ideal helpful response and maximize the reward margin between $y\_{\tau(1)}$ and “hard” dispreferred responses, prioritizing the minimization of false negatives.
>
> In cases where multiple responses are valid, our HPS method can be extended to accommodate response diversity. Specifically, we can formulate a weighted HPS loss, treating each valid response as a preferred one in its respective loss term. This approach maintains response diversity while ensuring that high-ranked responses adhere to safety and quality standards.
>
> For instance, given a training sample $d=(x,y\_{\tau(1)},y\_{\tau(2)},\dots,y\_{\tau(n)})\sim\mathcal{D}$, if both $y\_{\tau(1)}$ and $y\_{\tau(2)}$ are helpful responses, we can redefine the objective to train the model to reject all dispreferred and potentially harmful responses $(y\_{\tau(i)})\_{i=3}^n$, ensuring that it generates only the preferred responses $y\_{\tau(1)} $ and $ y\_{\tau(2)}$ for a given prompt $x$. The modified loss function is defined as a weighted sum of two HPS losses:$$\mathcal{L}\_{\boldsymbol{\theta}}=\mathcal{L}\_1+\lambda\cdot\mathcal{L}\_2$$where $\lambda$ is a weighting hyperparameter, and$$\mathcal{L}\_{1}=\mathbb{E}\_{d\sim\mathcal{D}}-\log\left(\frac{e^{{r\_{\theta}(x,y\_{\tau(1)})}}}{e^{{r\_{\theta}(x,y\_{\tau(1)})}}+ N\_{1}\cdot\mathbb{E}\_{y\sim p(y)}[e^{{r\_{\theta}(x,y)}}q\_{1}(x,y)]}\right),$$$$\mathcal{L}\_{2}=\mathbb{E}\_{d\sim\mathcal{D}}-\log\left(\frac{e^{{r\_{\theta}(x,y\_{\tau(2)})}}}{e^{{r\_{\theta}(x,y\_{\tau(2)})}}+ N\_{2}\cdot\mathbb{E}\_{y\sim p(y)}[e^{{r\_{\theta}(x,y)}}q\_{2}(x,y)]}\right),$$with$$q\_{1}(x,y)=\frac{e^{\gamma\cdot r\_{est}\left(x,y\right)}}{\sum\_{i=2}^{n}e^{\gamma\cdot r\_{est}\left(x,y\_{\tau(i)}\right)}},$$$$q\_{2}(x,y)=\frac{e^{\gamma\cdot r\_{est}\left(x,y\right)}}{\sum\_{i=3}^{n}e^{\gamma\cdot r\_{est}\left(x,y\_{\tau(i)}\right)}},$$$N\_{1}=n-1$, $N\_{2}=n-2$, and $p(y)$ is the probability distribution of the dispreferred response $y$. By optimizing the weighted HPS loss $\mathcal{L}\_{\boldsymbol{\theta}}$, the model is encouraged to rank $y\_{\tau(1)}$ and $y\_{\tau(2)}$ above all dispreferred and potentially harmful responses $(y\_{\tau(i)})\_{i=3}^n$, thereby maintaining both helpfulness and response diversity.
>
> We will include this discussion in the revision.

---

### Official Review · Reviewer_v4s5 · 2025-03-14

**Overall Recommendation:** 2

**Summary:**

The paper introduces **Hard Preference Sampling (HPS)**, a framework for aligning Large Language Models (LLMs) with human preferences. It addresses issues in existing methods (PlackettLuce and Bradley-Terry models) by prioritizing preferred responses, explicitly rejecting dispreferred/harmful ones, and focusing on "hard" dispreferred responses to enhance rejection. HPS uses single-sample Monte Carlo sampling for efficiency and maximizes reward margins for clearer distinctions. Experiments on HH-RLHF and PKU-Safety datasets show HPS achieves comparable BLEU/reward scores while improving reward margins and reducing harmful content.

**Claims And Evidence:**

HPS improves upon PL, but the experiments only include two responses with human-rated preferences, requiring scenarios with n > 2.

**Essential References Not Discussed:**

No

**Experimental Designs Or Analyses:**

Yes

**Methods And Evaluation Criteria:**

In many scenarios, it is not necessary for the less preferred option to be rejected; it is sufficient for the preferred option to be ranked higher than the less preferred one.

**Other Comments Or Suggestions:**

The subscript is missing in lin244 formula.

**Other Strengths And Weaknesses:**

Please see questions.

**Questions For Authors:**

• When n=2, iPL simplifies to BT, so it would be helpful to understand why the results of DPO-PL and DPO-BT in Table 2 show notable differences.
• Since DPO is sensitive to the beta parameter, a more comprehensive comparison could involve testing different beta values, plotting KL divergence on the x-axis and performance metrics on the y-axis, to better assess the effectiveness of DPO-BT and DPO-HPS.

**Relation To Broader Scientific Literature:**

Yes

**Theoretical Claims:**

No

---

> ### Author Rebuttal · Authors · 2025-04-01
>
> Thank you for the insightful and valuable comments! In the following, we provide our point-by-point response and hope our response helps address your concerns. We also look forward to the subsequent discussion which may further help solve the current issues.
>
> **1) PL simplifies to BT when $n=2$, but differs when $n \geq 3$**, explaining their different results in Table 2. In this experiment, each prompt has 100 responses ($n=100$). BT selects the most preferred and dispreferred responses for training, whereas PL uses all 100 responses to compute its loss (Eq. 4 and 5), leading to different performance outcomes.
>
> **2) Regarding the sensitivity of DPO**, to address your concern, we have conducted experiments with $\beta=(0.1,0.25,0.5,0.75,1)$ and report the KL divergence $\mathbb{D}\_{KL}[\pi_{\theta}(y_w|x)||\pi_{ref}(y_w|x)]$ across these values, where $x$ is the prompt and $y_w$ is the winning response in the test set. The results in Tab.1 demonstrate the superiority of our HPS: it achieves the highest $RM_{R-DPO}$ for all KL values, confirming that HPS leads to stronger rejection of harmful responses.
>
> ### Tab.1 Ablation results with $\beta$ on HH-RLHF under fine-tuning setting.
> |$\beta$|Method|KL|BLEU|Reward|$RM_{DPO}$|$RM_{R-DPO}$|
> |-|-|-|-|-|-|-|
> |0.1|DPO-BT|8.463|0.230|0.431|0.349|-0.455|
> |0.1|DPO-HPS|11.767|0.232|0.430|2.723|2.040|
> |0.25|DPO-BT|5.888|0.231|0.431|-0.206|-1.188|
> |0.25|DPO-HPS|6.972|0.230|0.431|-0.146|-0.828|
> |0.5|DPO-BT|2.661|0.229|0.430|-0.239|-1.022|
> |0.5|DPO-HPS|3.091|0.227|0.428|-0.228|-0.911|
> |0.75|DPO-BT|2.996|0.225|0.428|-0.264|-1.046|
> |0.75|DPO-HPS|2.192|0.226|0.427|-0.242|-0.925|
> |1|DPO-BT|2.043|0.227|0.430|-0.308|-1.990|
> |1|DPO-HPS|2.015|0.225|0.429|-0.316|-1.178|
>
> Moreover, we conducted a user study by selecting 15 prompt questions from HH-RLHF and 15 from PKU-SafeRLHF. For each question, four responses generated by SFT, DPO-BT, DPO-PL, and DPO-HPS are rated by 20 human evaluators on a 1–5 scale. To avoid bias, models were anonymized, and the response order was randomized. As shown in Tab.2, HPS achieves the highest quality score among all methods.
>
> ### Tab.2 Human evaluation on user study dataset.
> |Method|Quality Score|
> |-|-|
> |SFT|3.63|
> |DPO-BT|3.82|
> |DPO-PL|3.69|
> |DPO-HPS|3.93|
>
> **3 For experiments)**, Tables 2–4 in our paper demonstrate that HPS-based methods consistently outperform other methods. For each prompt, DPO-BT only selects the most preferred and dispreferred responses among $n=100$ responses for training, while DPO-PL and DPO-HPS use all $n=100$ responses.
>
> Moreover, Table 5 presents an ablation study analyzing the impact of varying the number of responses $n=5,20,50,100$ on preference optimization. The results indicate that DPO-HPS scales better and achieves superior preference optimization with larger response sizes.
>
> Specifically, we follow (arXiv:2306.17492) and expand response data by generating 100 responses using *RLHFlow/Llama3-v2-DPO* (arXiv:2405.07863) per prompt. The corresponding rewards are computed via *Skywork-Reward* (arXiv:2410.18451). Then we use methods like DPO-BT, DPO-PL, and DPO-HPS to fine-tune the language model on these data. Our preference fine-tuning methods explicitly leverage this broader response set rather than being constrained to the $n=2$ case.
>
> **4)**,
> Existing LLMs generate a single response autoregressively, which means that there is no inherent mechanism to ensure whether the generated content is harmful or harmless. This limitation raises concerns in scenarios that require adherence to safety and ethical standards.
>
> Additionally, if we generate $K$ responses using the LLM, the computational cost becomes substantial due to the inference overhead. Even with multiple generated responses, LLMs without refined preference alignment cannot rank these responses autonomously, making it difficult to identify the top-ranked response based on quality.
>
> To select the best response, a well-trained reward model is needed to rank the generated responses based on their quality and choose the highest-reward one. However, this approach introduces two key limitations:
> - Inference Cost: Generating $K$ responses incurs significant computational overhead.
> - Safety Concerns: While the top-ranked response may be of high quality, it is not guaranteed to be harmless, as both the LLM and reward model may fail to capture all potential risks.
>
> To address these issues, we propose HPS, which ensures that lower-ranked responses with undesirable content are minimized, prioritizing the reduction of false negatives (L138). This consideration is particularly crucial in applications requiring high-quality and safe content generation, such as healthcare and education.
>
> Furthermore, our HPS can be extended to the setting where multiple top responses are valid. Please see our response to Reviewer 674W.
>
> **5) For the missing subscript**, we will correct it, and also carefully review the entire manuscript. Many thanks for your thorough proofreading and effort!

---

### Official Review · Reviewer_xe8N · 2025-03-21

**Overall Recommendation:** 4

**Summary:**

This work introduces **Hard Preference Sampling**, a framework for aligning large language models to human preferences. HPS introduces a training loss that adaptively penalizes dispreferred responses, and focuses on “hard” dispreferred responses, i.e. responses that are similar to preferred responses to increase reward margins. HPS is also as efficient as Bradley-Terry models during training, as for each preferred response, it samples a single dispreferred response. The authors also show theoretical bounds on sample complexity that scale better than Plackett-Luce models, as well as on reward margin quality. Finally, this work also empirically demonstrates positive win rates with LLMs as a judge compared to baselines on popular preference datasets for helpfulness, harmlessness, and safety.

**Claims And Evidence:**

**Claim 1** : model can distinguish between preferred and highly similar dispreferred responses more effectively – the authors empirically show better reward margins compared to baselines on HH-RLHF and PKU-Safety (Tables 2 - 5) and error bounds on optimal solutions in Thm 1. It is however unclear if the margin is between two responses that are similar, and if so, how they are similar (what is used to measure similarity?). This should be explicitly specified.

**Claim 2** : HPS provably improves sample complexity over vanilla PL loss – the authors show this in Thm 1.

**Claim 3** : HPS provably maximizes the reward margin for any prompt – the authors show this in Thm 2 and empirically in Tab 2 and Tab 4.

**Essential References Not Discussed:**

To the reviewer’s knowledge, no essential reference relating to Direct Alignment Algorithms has been overlooked by this work. The reviewer does recommend a (optional) discussion comparing to popular explicit reward modeling methods such as PPO (https://arxiv.org/abs/1707.06347) and GRPO (https://arxiv.org/abs/2402.03300), which would better situate this work in the literature. This can be done in the Appendix.

**Experimental Designs Or Analyses:**

The overall goals and high-level evaluation criteria makes sense (see comments in “Methods and Evaluation Criteria”), but the setting is quite unclear and underspecified in the writeup. The reviewer is left with several questions, all of which need to be explicitly mentioned in the revision:
1. Practically, what value of $\gamma$ is used ? Since the penalty for the dispreferred response (and improvements in reward margins) seem to rely on this, it is important to disclose.
2. Practically, what is used to compute $r_{est}$, which itself is used to compute $q(x,y)$ (L 243 - 244) ?
3. What exactly is the experimental setting with HH-RLHF and PKU Safety? In the reviewers understanding, in Tab 2 “DPO-PL” is a Llama3-8B base model trained on HH-RLHF with DPO assuming a Plackett-Luce preference model. Similarly “IPO-HPS” is a Llama3-8B base model trained on HH-RLHF with IPO assuming a HPS preference model. Is this understanding correct? This should be made a little more clear in the “Baselines” paragraph in Section 6
4. The authors mention that prompts from these datasets used to generate 100 responses with each Llama3 model (SFT, PL, BT, HPS) which are then scored by the top-10 safety ranked RM (L 312) – is this understanding correct? Further, the reward margins are computed over only 2 responses (Tab 1) – how then are these 100 responses used? In the “Implementation” paragraph (Sec 6), the authors state that $n=5$ responses are used for PL methods, is this only for win rates? (L370). This is quite confusing.

**Methods And Evaluation Criteria:**

The proposed method is to compare HPS to status quo preference models (BT and PL) on popular safety datasets – Anthropic’s Helpful-Harmless and PKU Safety. They evaluate their method with an LLM as a judge to approximate real human preferences, i.e. win rates compared to baselines. They also evaluate transfer learning, i.e. training on one dataset and evaluating on another, which is a relevant and important measure of robustness. This criteria is sound, but relies on the base judge model’s quality (Qwen-2.5 Instruct). Ideally, there needs to be a survey with real human participants.

**Other Comments Or Suggestions:**

One optional suggestion: changing the acronym of this work. There is a pre-existing popular work in preference learning / alignment also called HPS (https://arxiv.org/abs/2306.09341), which is often used to score text-to-image models like Stable Diffusion or DALL-E. It may be slightly confusing to the community to also refer to this work as "HPS".

Misc comments
1. The point discussing that BT models "leave other problematic responses unaddressed" (L207- 210) is unsubstantiated. Popular state-of-the-art LLMs use BT models and work very well at scale while addressing these other problematic responses. This ties in to my request for a distinction of HPS from BT (see "Relation To Broader Scientific Literature" section).
2. The point that PL loss trains models "without considering the inter-ranking relationship among dispreferred responses" (L219) is not strictly true. Each dispreferred response becomes the preferred response for the adjacent response which is slightly more dispreferred, and each of these pairwise losses are summed up. Thus, the most dispreferred response (last in the PL ranking) is a part of every loss $\mathcal{L}_j$ and is weighted more highly as it is considered multiple times. What is true is that the *weights* for each loss are typically the same (1). I recommend rephrasing this portion to make this clearer

**Other Strengths And Weaknesses:**

Overall, **I like this work**. It shows both empirically (on HH-RLHF and PKU-Safety) and theoretically (Thm 1- 3) that dispreferred responses can be "pushed away" from preferred responses with a simple objective modification in a sample efficient manner. I think it addresses a relevant problem: differentiating between dispreferred responses (i.e. all dispreferred responses are not equally bad) while maintaining low sample complexity. I would also like to highlight some weaknesses with the current version of the manuscript than can be improved to make a much stronger version of the work.

**Weaknesses**:
1. How is this work different from weighting the dispreferred response in a Bradley-Terry model according to its quality? (see "Relation To Broader Scientific Literature" for details)
2. How is the "importance-weighted dispreferred response" that is the backbone for HPS sample efficiency (L262) chosen? This is very important to describe in detail as it is a crucial part of the algorithm.
3. The experiment setting needs better clarity (see "Experimental Designs Or Analyses" section for details)

With these strengths and weaknesses in mind, I currently recommend a weak accept (3). If these three primary weaknesses are addressed in the revision, I am willing to increase my score to accept (4).

Edit: As the above weaknesses have been discussed and committed to be updated in the rebuttal, I update my score to accept (4).

**Questions For Authors:**

I had several questions about the evaluation setup (see "Experimental Designs Or Analyses" section)

**Relation To Broader Scientific Literature:**

This work is relevant to the broader scientific literature. Empirically, it compares to many state-of-the-art and popular preference tuning methods, including DPO, IPO, EXO, SPPO, and NCA. It also addresses an important problem, i.e. the relative quality or "badness" of a dispreferred response explicitly, whereas Plackett-Luce models do so implicitly via a pairwise ranking across all pairs (thus creating an ordering over all responses). The authors do show favorable results compared to baselines in terms of reward margins between preferred and dispreferred responses, but without a qualitative human study, it is difficult to directly compare dispreferred responses across methods (e.g. HPS vs BT). From HPS’ “importance-weighted sampling” for efficiency (L264), the reviewer is also unclear how HPS differs a Bradley-Terry model where the dispreferred response is quality-weighted in some sense (e.g. with the same score obtained from a top-10 safety ranked reward model used for win rate). A brief discussion on the differences between HPS and quality-weighted BT would better highlight the contribution of this work.

**Theoretical Claims:**

I read through the theoretical claims, but I did not check for correctness of proofs.

---

> ### Author Rebuttal · Authors · 2025-04-01
>
> Thank you for the insightful and positive comments! We provide our response and hope our response addresses your concerns. We also look forward to the subsequent discussion which may further help solve the current issues.
>
> **1) For quality-weighted BT**, we could not find prior work directly related to it but identified two relevant methods: WPO (arXiv:2406.11827) and LiPO-$\lambda$ (arXiv:2402.01878). We will discuss them first, and will provide further discussion if you can provide specific references.
>
> In **WPO**, weights are assigned to response pairs based on their occurrence probability:
> $$\mathcal{L}_{WPO}=-\mathbb{E}\_{(x,y_w,y_l)\sim\mathcal{D}}[w(x,y_w)w(x,y_l)\log p(y_w\succ y_l|x)],$$where $w(x,y)=\exp{\frac{1}{|y|}\sum^{|y|}\_{t=1}\log\pi\_{\theta}(y_t|x,y\_{<t})}$ and $|y|$ is the number of tokens in the output. So WPO modifies response weights to better align with on-policy data while following DPO to consider only the most preferred and most dispreferred responses. In contrast, our HPS accounts for multiple responses and focuses on leveraging the varying informativeness of all dispreferred responses.
>
> Discussion of **LiPO-$\lambda$** can be found in our response **1)** to Reviewer 85yJ.
>
> **2) Regarding importance-weighted dispreferred response**, we directly sample one dispreferred response according to the importance-weighted distribution (L244):$$q(x,y)=\frac{e^{\gamma\cdot r_{est}(x,y)}}{\sum_{i=2}^{n}e^{\gamma\cdot r_{est}(x,y_{\tau(i)})}},$$where $(y_{\tau(i)})_{i=2}^{n}$ denote the dispreferred responses of a prompt $x$. Thus, harder dispreferred responses will be sampled with higher probability and contribute more to the loss due to their higher probability $q(x,y)$. Then, we can incorporate the sampled dispreferred response $y$ into the loss function Eq. 9 for training.
>
> **3) For response similarity**, it refers to two responses having comparable rewards, indicating shared semantics (key information). For example, Fig. 1 in submission shows that for a given prompt $x$, response $y_{\tau(1)}$ is more similar to $y_{\tau(2)}$ than to $y_{\tau(3)}$ since their content is closer. Consequently, $y_{\tau(1)}$ and $y_{\tau(2)}$ receive similar rewards, reinforcing their similarity.
>
> However, we may not fully understand the question—please clarify if needed.
>
> **4) For human evaluation**, we conducted a user study with human participants. See Tab.2 in our response **2)** to Reviewer v4s5 for details.
>
> **5) Regarding reward $r_{est}$ of each ranked response $y_{\tau(i)}$**, we either use the given reward $r_{est}$ or estimate it with a pretrained preference-aligned reward model (L230). In our experiments, we use Skywork-Reward-Llama-3 (arXiv:2410.18451). The scaling factor $\gamma$ is linearly increased from -5 to 5 at every 20\% interval of the training process.
>
> **6) For experimental setting**, your understanding is correct. In Table 2, DPO-PL and IPO-HPS independently fine-tune Llama3-8B on HH-RLHF using DPO with a PL preference model and IPO assuming an HPS preference model.
>
> **7) For experiments**, we follow (arXiv:2306.17492) and expand response data by generating 100 responses using *RLHFlow/Llama3-v2-DPO* per prompt. The corresponding rewards are computed via *Skywork-Reward-Llama-3*. Then we use methods like DPO-BT, DPO-PL, and DPO-HPS to fine-tune the language model on these data. For each prompt, DPO-BT only selects the most preferred and dispreferred responses for training, while DPO-PL and DPO-HPS use all 100 responses.
>
> To evaluate alignment, we measure Reward Margins (RM) in Table 1, where higher RM scores indicate better preference alignment with minimal harmful or biased outputs.
>
> For PL methods like DPO-PL, directly using 100 responses per prompt (n=100) incurs excessive GPU memory costs (see Eq. 9 $\mathcal{L}_{PL}$). To mitigate this, we reformulate the PL sub-loss $\mathcal{L}\_{j}(d)$ using Monte Carlo sampling:
> $$
> \mathcal{L}\_{j}(d)=-\log\left(\frac{e^{r\_{\boldsymbol{\theta}} (x,y\_{\tau(j)})}}{e^{r\_{\boldsymbol{\theta}}(x,y\_{\tau(j)})}+N \cdot\mathbb{E}\_{y\sim u\_{j}(x,y)}[e^{r\_{\boldsymbol{\theta}}(x,y)}]}\right),
> $$
> where $N=n-j$ and $u\_{j}(x,y)$ is a uniform distribution over dispreferred responses. Instead of using all $N$ dispreferred responses, we sample 5 per loss term $\mathcal{L}\_{j}$, which is the maximum our 4×L40S GPUs can accommodate.
>
> This sampling-based PL formulation is theoretically equivalent to vanilla PL and does not impact performance. The table below confirms that randomly sampling 5 or 1 dispreferred response from 100 yields similar performance on HH-RLHF and PKU-SafeRLHF. Since the strategy is developed in this work and used by HPS, it ensures a fair comparison between PL and HPS.
>
> |Dataset|BLEU|Reward|$RM_{DPO}$|$RM_{R-DPO}$|
> |-|-|-|-|-|
> |HH-RLHF|0.231|0.430|-0.859|-1.480|
> |PKU-SafeRLHF|0.302|0.410|-5.804|-6.061|
>
> **8)** We have briefly discussed explicit Preference Fine-Tuning methods (L155). We will discuss them more in the Appendix.

---

> > ### Comment · Reviewer_xe8N · 2025-04-04
> >
> > I thank the authors for their response! Here are a few comments
> >
> > 1.  **The relation to importance-weighting samples with BT model**: I appreciate the reference to two prior works discussing importance-weighting a preference pair with BT models. Please add these in the revision. A "harder" response is determined entirely by its estimated reward $r_{est}$ (L244) and thus how  $r_{est}$ is computed is crucial. I agree with reviewer 85yJ that this and details about the sampling strategy is still a little unclear ( see point 2 below). Please explicitly add a discussion in the revision with the details in your rebuttal to reviewer 85yJ (*3) Sampling a single importance-weighted dispreferred response DOES NOT reduce HPS to standard BT*), as this point was also unclear to me (hence my question about HPS being equivalent to importance-weighted BT).
> >
> > 2. **Regarding $r_{est}$**: the authors mention ``directly access its reward $r_{est}$ if available in the dataset D'' (L229-230). What does "directly access" mean? In the rebuttal point 5, the authors mention "the given reward $r_{est}$" - from where is this score given? The details about using Skywork-Reward-Llama-3 and practical details about $\gamma$ must be reported explicitly in the writeup revision.
> >
> > 3. **Response similarity**: it was unclear how similarity was computed in the original manuscript. Through the rebuttal it is now clear this means that the scalar reward is $\epsilon$-close for some small $\epsilon$. This should be made explicit in the writeup revision.
> >
> > 4. **Clarity of Experimental Section writeup**: I thank the authors for their rebuttal clarification. Please directly include these details (Rebuttal point 6 and 7) in the experiments section in the revision, since the setup was still a little confusing until I read the rebuttal.
> >
> > 4. **User study**: I appreciate the inclusion of a user study, this supports the claim that HPS does better than baselines. However, the authors must provide much more detail about this study - what exactly where participants asked? What does "Quality Score" mean? What did each score on the Likert scale correspond to in the instruction (e.g. 1 - bad, 5 - good)
> >
> > With the above changes in the revision, I will update my score to 4 (accept).

---

> > > ### Author Response · Authors · 2025-04-06
> > >
> > > Thank you for the detailed comments. Please kindly see below for our responses to your comments:
> > >
> > > **1)** By “directly access,” we mean that if a scalar reward is explicitly annotated (i.e., 'given') for each response in the dataset $\mathcal{D}$, we could use the value as $r\_{est}$ without requiring any further inference or estimation.
> > >
> > > **2)** In our designed user study, the “Quality Score” refers to the quality and helpfulness of the generated response. Participants rated responses using a 5-point Likert scale, where 1 indicates poor quality and 5 indicates high quality. We will provide details of the user study, including the instructions given to participants and the evaluation criteria, in our paper.
> > >
> > > We will also revise the Method and Experimental sections to improve the paper’s clarity. Thank you again for your constructive feedback!

---

### Official Review · Reviewer_SuNZ · 2025-03-23

**Overall Recommendation:** 4

**Summary:**

This paper adapts the concept of hard negative sampling, which was previously employed in metric learning and contrastive learning settings, to preference alignment. Hard Preference Sampling (HPS) framework reconsiders the loss function derived by incorporating reward into the  Plackett-Luce (PL) model and uses a modified version with following contributions:

- Training loss in HPS framework boosts hard negatives$-$dispreferred responses with high rewards.

- They sample only one negative response using Monte Carlo, which reduces training costs, and it appears to work well while maintaining alignment quality.
- The paper claims that optimizing the HPS loss maximizes the reward margin and HPS provably improves sample complexity over the vanilla PL loss.

**Claims And Evidence:**

I have couple of clarification questions (See below).

**Essential References Not Discussed:**

No, up to my understanding.

**Experimental Designs Or Analyses:**

They are convincing.

**Methods And Evaluation Criteria:**

Yes.

**Other Comments Or Suggestions:**

No.

**Other Strengths And Weaknesses:**

**Strengths:**
- It is a useful adaptation of ideas to preference alignment setting to improve safety and reliability of LLMs.
- I find the paper well written and easy to follow.

**Weaknesses:**
- Hard negative sampling is a known technique employed in various settings, such as metric learning and contrastive learning. The maximum margin property of hard negative sampling has been demonstrated in these contexts. Therefore, the originality of the paper lies in applying this technique to preference alignment, which limits the technical contributions and makes the conceptual contributions more significant in highlighting the distinguishing characteristics.

**Questions For Authors:**

Can you elaborate on what leads to the difference in sample complexity bounds, which differ by an $n$ factor?

**Relation To Broader Scientific Literature:**

The paper proposes Hard Preference Sampling framework for better alignment quality. This work can be considered as an adaptation of hard negative sampling technique from metric learning and contrastive learning literature to preference alignment. The vanilla model uses the loss function derived by incorporating reward into the  Plackett-Luce (PL) model. This paper reconsiders this PL model based loss function. In a broad sense, In a broader sense, their results provide intuition for safer and more responsible language models.

**Theoretical Claims:**

I wasn't able to check all the details to fully confirm its mathematical correctness, but I don't see any problems.

---

> ### Author Rebuttal · Authors · 2025-04-01
>
> Thank you for the insightful and positive comments! In the following, we provide our point-by-point response and hope our response helps address your concerns. We also look forward to the subsequent discussion which may further help solve the current issues.
>
> **1) For hard negative sampling**, this work is the first to extend hard negative sampling to preference alignment, addressing new task-specific challenges with novel and effective solutions:
>
> **a) Handling Varying Informativeness of Dispreferred Responses.** In metric and contrastive learning (arXiv:2010.04592, arXiv:2108.09335), hard negatives are typically selected based on representation similarity to a positive anchor. However, in RLHF, where responses are generated autoregressively, obtaining effective sentence embeddings is impractical. Instead, we define "hardness" in the reward space, where dispreferred responses similar to preferred ones (i.e., with close reward scores) are considered harder (L252). Furthermore, selecting hard negatives using $q(x,y)$ presents an intractable distribution challenge (L234), which we address via Monte Carlo importance sampling (Eq. 9).
>
> **b) Improving Sampling Efficiency.** In previous work on metric and contrastive learning, backbone models such as ResNet or GoogLeNet were employed in vision tasks, where the model size is approximately $1\%$ that of Llama-3-8B in the RLHF setting. To ensure computational efficiency, we reformulate HPS into an efficient sampling approach, using a single Monte Carlo sampling to select a single dispreferred response per training sample.
>
> **c) Theoretical Analysis of Hard Negative Sampling in Preference Alignment.** Our work is the first to provide a theoretical analysis of hard negative sampling in this context, offering new insights into alignment. We compare sample efficiency between the preference loss (PL) and HPS loss, demonstrating that HPS improves sample efficiency, particularly in data-scarce settings or when rapid convergence is needed. Additionally, we analyze how training with HPS maximizes the reward margin between preferred and hard dispreferred responses, ensuring a robust distinction between them. This strengthens alignment performance while minimizing undesired outputs. We will further elaborate on these distinctions in Sec. 1 and 4.
>
> **2) Regarding the differences in sample complexity bounds of our HPS and the PL**, intuitively, they stem from the structural distinction of two losses. The PL loss is composed of a summation of $n$ piece of one-to-$N$ contrast losses $\mathcal{L}\_{j}(d)$:$$\mathcal{L}\_{PL}=\mathbb{E}\_{d\sim\mathcal{D}}\sum\_{j=1}^{n}\mathcal{L}\_{j}(d)=\mathbb{E}\_{d\sim\mathcal{D}}\sum\_{j=1}^{n}-\log({e^{r\_{\theta} (x,y\_{\tau(j)})}/\sum\_{k=j}^{n}e^{r\_{\theta}(x,y\_{\tau(k)})}}),\tag{1}$$In contrast, our proposed HPS loss focuses on encouraging the model to rank the most preferred response $y\_{\tau(1)}$ against all other dispreferred responses $(y\_{\tau(i)})\_{i=2}^n$:$$\mathcal{L}\_{\theta}=\mathbb{E}\_{d\sim\mathcal{D}}-\log(\frac{e^{{r\_{\theta}(x,y\_{\tau(1)})}}}{e^{{r\_{\theta}(x,y\_{\tau(1)})}}+N\cdot\mathbb{E}\_{y\sim p(y)}[e^{{r\_{\theta}(x,y)}}q(x,y)]})\tag{2}$$with$$q(x,y)=\frac{e^{\gamma\cdot r\_{est}\left(x,y\right)}}{\sum\_{i=2}^{n}e^{\gamma\cdot r\_{est}\left(x,y\_{\tau(i)}\right)}}.\tag{3}$$The HPS loss $\mathcal{L}\_{\theta}$ only uses one component, $\mathcal{L}\_1$, from the full summation in $\mathcal{L}\_{PL}$. Thus, the structural distinction between the two loss functions leads to the $n$-factor discrepancy in the asymptotic error bound.
>
> More specifically, as shown in Appendix B.1, the difference between the HPS loss $\mathcal{L}\_{\theta}$ and the PL loss $\mathcal{L}\_{PL}$ has a direct impact on their gradients. We follow the mathematical notations in our paper. Specifically:
>
> - For HPS-based loss:$$\|\nabla\mathcal{L}\_{\theta}(\theta^{*})\|^{2}\_{\Sigma^{-1}\_{\mathcal{D}}}\leq C\cdot\frac{d+\ln(\frac{1}{\delta})}{m}$$
> - For PL-based loss:$$\|\nabla\mathcal{L}\_{PL}(\theta^{*})\|^{2}\_{\Sigma^{-1}\_{\mathcal{D}}}\leq C n^{4} \cdot \frac{d+\ln(\frac{1}{\delta})}{m}$$
>
> The error bound for the HPS-based method $\|\Delta\_{HPS}\|\_{\Sigma\_{\mathcal{D}}}$, where $\Delta\_{HPS}:=\theta\_{HPS}-\theta^{*}$, is bounded by
>
> $\frac{\|\nabla\mathcal{L}\_{\theta}(\theta^{*})\|\_{\Sigma^{-1}\_{\mathcal{D}}}}{\zeta}$
>
> with$$\zeta = \frac{1}{2+\exp(2\alpha\_{0}+\ln(n-1))+\exp(-2\alpha\_{0})}$$as stated in Thm. 1. Similarly, the error bound for the PL-based method $\|\Delta\_{PL}\|\_{\Sigma\_{\mathcal{D}}}$, where $\Delta\_{PL}:=\theta\_{PL}-\theta^{*}$, is bounded by
>
> $\|\nabla\mathcal{L}\_{PL}(\theta^{*})\|\_{\Sigma^{-1}\_{\mathcal{D}}}.$
>
> From this, we observe that $\Delta\_{HPS}$ exhibits an error bound of $\mathcal{O}(n)$, whereas $\Delta\_{PL}$ has an error bound of $\mathcal{O}(n^2)$, differing by a factor of $n$. We will integrate this intuitive explanation into Thm. 1 for clarity.

---

### Decision · Program_Chairs · 2025-05-01

**Decision:**

Accept (poster)

**Comment:**

This paper focuses on preference based safety alignment for and proposes sampling hard preferences (HPS). The analysis presented in the paper relies on the assumption data being generated from the true model and does not address model mismatch. As reviewers have pointed out, while the idea of hard negative sampling exists in related literature of metric learning and contrastive learning, application of this idea to preference based alignment setting is interesting. The theoretical analysis assumes that data comes from the true model which while limiting is still an interesting analysis providing insights on why HPS is advantageous over using the ranked list under PL model. In summary the strengths outweigh the limitations.

Please include the discussion to the additional relevant literature pointed out the by the reviewers and the promised discussion during rebuttal-response for weaknesses raised.